# The Ranking Trick: A Simple and Robust Alternative to Score-Based Regression for AutoML

**Hernán C. Vázquez**[1]  **Jorge A. Sanchez**[1]  **Veronica Bogado**[1]  **Tobias Pucci Romero**[1]

[1]MercadoLibre, Inc

**Abstract**   Traditional approaches to pipeline selection in automated machine learning (AutoML) typically rely on predicting the absolute or relative performance scores of candidate pipelines for a given task, based on data acquired from previous tasks—i.e., meta-learning. This process can be complex due to the need for task-specific regression models and performance metrics. In contrast, rank-based methods estimate the relative ordering of pipelines, which aligns more directly with the decision-making nature of the selection task. Although ranking-based approaches have been explored previously, prior work often relies on computationally expensive pairwise comparisons or complex listwise formulations. In this study, we adopt a simpler alternative: reformulating the prediction target from absolute scores to rank positions—without modifying model architectures. This "ranking trick" enables the use of regression models while leveraging positional information. It is general and compatible with a wide range of existing AutoML techniques. Additionally, through controlled experiments, we show that these rank-based regression models are significantly less sensitive to noisy or overfitted meta-learning data, a common issue in practical AutoML settings. As a result, this approach enables more robust, metric-agnostic solutions and facilitates evaluation through ranking metrics such as NDCG and MRR. We evaluate this formulation across three large-scale OpenML benchmarks, demonstrating consistent advantages for ranking-based regression models. Furthermore, we explore its integration with Bayesian optimization and Monte Carlo Tree Search, yielding improved results in ranking quality. Finally, we identify a strong relationship between ranking-based metrics and key AutoML objectives such as final performance score and time-to-solution, providing guidance for AutoML practitioners.

## 1 Introduction

Machine learning (ML) has become an essential component of countless scientific and industrial endeavors, ranging from healthcare (Waring et al., 2020) to e-commerce (Micu et al., 2019). However, applying machine learning in the real world is far from straightforward, as it involves multiple steps such as data preparation (preprocessing), feature and model selection, hyperparameter tuning, performance evaluation, etc. (Barbudo et al., 2023; Vazquez, 2022). The complexity in the design of machine-learning solutions led to the development of automated Machine Learning (AutoML) techniques that seek to relieve ML practitioners from repetitive and time-consuming tasks that have a large impact on the final performance of ML systems (Karmaker et al., 2021).

While AutoML helped widen the adoption of machine learning solutions at scale by simplifying the process of finding optimal configurations, it faces significant challenges. One such challenge is efficiently leveraging past experiences to improve performance on new tasks. This problem led to the development of meta-learning algorithms and techniques (Vanschoren, 2019) to capture the knowledge gained from solving a variety of tasks to help find optimal solutions for novel (future) ones. Meta-learning algorithms rely on meta-models to predict the performance of a given configuration (pipeline) on a specific dataset. This prediction serves as an approximation of the actual performance that the model would demonstrate if it were trained with such data. The benefit of this approach is that it prevents from actually training the model on the data in question, saving

time and costs. Note that predicting pipeline performance involves predicting specific metrics for the different types of ML tasks (classification, regression, etc.), often requiring different models for each task-specific metric being optimized (Karmaker et al., 2021).

We can recognize two broad strategies in the design of an AutoML system. One is to build an optimal machine learning solution from a pool of simpler components (Ren et al., 2021), and the other is to select it from a set of predefined options or configurations (Yang et al., 2020). When the problem is about choosing among given solutions, it can be viewed as a ranking or recommendation task, where the goal is to rank the set of possible configurations according to their effectiveness in solving the task. However, in many cases, the efforts of AutoML systems are focused on predicting the performance of configurations rather than tackling the selection or decision-making problem directly. For instance, when Bayesian optimization is used (Hutter et al., 2011), the selection of configurations often involves choosing the next most promising configuration based on a surrogate model. This model is designed to capture the relationship between the configuration and the target metric, to inform the selection process about which configuration to evaluate next. We argue that the selection problem could be addressed more effectively by framing it as a decision-making problem, thereby facilitating a more direct approach to determining the most valuable solutions.

Previous works explored different approaches to navigate the complex combinatorial space of available options (models, preprocessing steps, hyperparameters, etc.) in AutoML. Drawing parallels to finance portfolio management, some methods (Fusi et al., 2018; Yang et al., 2019) optimize model and pipeline selection based on historical performance, akin to optimizing a collection of assets. Other methods (Laadan et al., 2020; Feurer et al., 2022), directly adopt ranking and meta-learning strategies, demonstrating improvements in the efficiency of the AutoML workflow through data-driven decision-making and the pre-ranking of machine learning pipelines. However, although these approaches make use of ranking information to speed up the search process and improve AutoML efficiency, they either do not formulate the underlying ranking problem explicitly, or do so through computationally expensive methods such as pairwise comparisons (Lindauer et al., 2015; Laadan et al., 2019; Kostovska et al., 2023).

This paper addresses these challenges by proposing a simple reformulation of the pipeline selection task in AutoML: replacing the prediction of performance scores with the prediction of rank positions. Instead of relying on complex learning-to-rank (LTR) methods (Liu et al., 2009), we show that this "ranking trick" can be implemented with standard regression models by simply reformulating the prediction target as rank positions rather than absolute scores. This approach aligns more closely with the decision-making nature of the selection task and enables more robust, metric-agnostic solutions. We evaluate both score-based and rank-based variants of common selection strategies using ranking metrics such as NDCG and MRR. Furthermore, we explore how this reformulation can be integrated into classical sequential optimization techniques like Bayesian Optimization and Monte Carlo Tree Search. Experiments on public benchmarks demonstrate consistent advantages for ranking-based regression models across diverse conditions.

The main contributions of this paper are:

- A practical formulation of pipeline selection in AutoML using rank-based regressions. This approach better aligns with the true objective of selecting the most promising pipeline, while enabling more robust and metric-agnostic solutions, thereby simplifying comparisons.

- A methodological shift toward evaluating AutoML strategies using ranking-based metrics such as NDCG and MRR. We further analyze the correlation between these ranking metrics and conventional AutoML objectives, such as performance improvement and time-to-solution, providing deeper insights into their practical relevance.

- Empirical evidence demonstrating that ranking-based approaches consistently outperform traditional score-based methods under various conditions, including noisy or limited meta-learning

data. Experiments on public datasets validate the effectiveness and robustness of the proposed framework in real-world scenarios.

## 2  Using ranking information for AutoML

To prevent AutoML systems from repeatedly tackling similar tasks starting from scratch each time, meta-learning methods leverage data acquired from previous experience (meta-learning data) to improve performance on novel tasks (represented with meta-features) (Vanschoren, 2019). In our case, this corresponds to optimizing the procedure by which we select optimal pipelines for a wide variety of problems. We propose to do so by using ranking information instead of task-specific performance scores. In this way, the process of pipeline selection becomes independent of the absolute scores observed in the meta-learning data as well as from the different performance values achieved on each task.

To better motivate our proposed ranking-based approach, we first briefly describe the traditional score-based formulation, followed by our proposed ranking-based alternative. We conclude by outlining how this formulation can be naturally integrated into sequential optimization strategies such as SMBO (Sequential model-based optimization) (Hutter et al., 2011).

**Score-Based Models**. Score-based models represent most traditional approaches in the AutoML frameworks. These models aim to predict the target metric of a problem, such as classification accuracy or mean regression error. The predicted score is then used to select the most promising configuration or pipeline. This aligns with the pointwise approach in LTR theory, where the input space consists of individual configurations, and the output space is the score of these configurations on the target task.

Score-based approach can be formalized as follows. Let $\mathcal{C} = \{C_1, C_2, \ldots, C_n\}$ represent the set of all possible configurations or pipelines in the AutoML search space. The goal is to predict the performance score of each configuration $C_i$ for a new given dataset $D$. To do so, we rely on a (parametric) score prediction model $f : \mathcal{C} \times \mathcal{D} \to \mathbb{R}$ that takes a candidate configuration $C_i$ and dataset $D$ as inputs and predicts a score $f(C_i, D)$ that relates to the predictive performance of $C_i$ on $D$. The score predicted by $f$ is specific to each task (accuracy, F1-score, regression errors, etc.). We assume we know the true value of this score for each pipeline-dataset combination in the meta-training dataset. We denote the ground-truth scores as $s(C_i, D)$. Training the meta-model is based on regression towards this metric by optimizing a suitable loss, such as the mean squared error of the scores, using a meta-training dataset (e.g. mean squeare error of the scores) consisting of triplets of the form $\{(C_i, D_j, s_{ij})\}$, with $C_i \in \mathcal{C}$, $D_j \in \mathcal{D}$, and $s_{ij} = s(C_i, D_j)$.

**Ranking-Based Models**. Ranking-based models focus on directly learning the ranking order of different configurations. These models are trained using explicit ranking information. The two primary formulations in this category are the pairwise and listwise approaches. Despite the advantages of these techniques, they are often computationally expensive, as they require processing and comparing multiple configurations simultaneously, which increases training complexity. By transforming the input and output spaces from scores into ranking positions, we can incorporate listwise ranking information in a simpler and more efficient way, avoiding the high computational costs typically associated with these methods.

In this simplification of the listwise approach, the input space corresponds to permutations of the set of all configurations $\mathcal{C}$. Instead of directly working with the set of all permutations of $\mathcal{C}$, we can think of a ranking function $h : \mathcal{C} \times \mathcal{D} \to \mathbb{R}^{|\mathcal{C}|}$ that takes as input the set of available configurations and a dataset $D$, and outputs a vector of scores of dimensionality $|\mathcal{C}|$, whose $i$-th entry corresponds to the relevance of configuration $C_i$ for solving the problem. Sorting the elements of this vector provides the permutation indices that rank the elements of $\mathcal{C}$ according to their relevance for $D$. Training such a model involves using the entire set of configurations $\mathcal{C}$ and their

rankings for different datasets. The model learns to predict these rankings as accurately as possible. A schematic overview of our ranking-based formulation and its relation to traditional score-based models is presented in Appendix A.

**Integrating LTR into SMBO.** This formulation of LTR can be naturally integrated into sequential optimization strategies to improve the efficiency of pipeline selection. By replacing traditional performance estimators with ranking-based models, frameworks such as Bayesian Optimization and Monte Carlo Tree Search can prioritize configurations more effectively, reducing evaluation costs and improving search quality. This integration enables a ranking-aware perspective across both model-based and model-free optimization methods.

## 3 Experimental Setup

**Models**. We evaluate five categories of pipeline selection models:

1. **Random Selection**: Random pipeline selection as a control baseline.
2. **Average Best (Avg):** Pipelines ranked by average score or average rank on training data.
3. **Regression Models**: Score or rank prediction using regressors, without hyperparameter tuning. Linear Regression (LR), Lasso, Ridge, Random Forest (RF), Gradient Boosting (GB) an Light Gradient Boosting Machine (LGBM).
4. **Sequential Optimization**: SMBO strategies using Bayesian Optimization (BO) and Monte Carlo Tree Search (MCTS), adapted to use score- or rank-based models.
5. **AutoFolio**: An algorithm selector using pairwise score comparisons, evaluated here with score- and rank-based variants.

Except for the Random baseline, all approaches were implemented in both score- and ranking-based variants. For BO, we initialize the search with a meta-learned surrogate model (regression of scores or ranks) and iteratively select pipelines using a greedy acquisition strategy. MCTS is used in a model-free setting, where the reward function is either average score or average rank, and pipelines are selected using UCB and greedy rollout as in (Vazquez et al., 2022). After each round, the pipeline selected is removed from the candidate pool. The final selection order is taken as the output ranking. Furthermore, AutoFolio was included to compare its pairwise-based formulation combined with our ranking approach. Both score and rank variants were tested using default configurations.

**Datasets**. We evaluate on three scenarios based on OpenML datasets (as shown in Table 1): (1) OPENML-WEKA-2017 (from ASLib) with 105 classification tasks and 30 pipelines; (2) AMLB 2023 Classification with 71 tasks and 2160 pipelines; (3) AMLB 2023 Regression with 33 tasks and 1485 pipelines. The AMLB scenarios were created following best practices for reproducibility and coverage of diverse learning problems. The full list of OpenML task IDs can be found in Appendix B.

| Scenario | #Tasks | #Pipelines | Metric |
|---|---|---|---|
| ASLib - OPENML-WEKA-2017 | 105 | 30 | Accuracy |
| AMLB Classification 2023 | 71 | 2160 | Balanced Accuracy |
| AMLB Regression 2023 | 33 | 1485 | Negative RMSE |

Table 1: Summary of the benchmark scenarios.

**Data Collection and Preprocessing**. Pipeline evaluations were executed using a grammar-based search space (Vazquez et al., 2022), with a one-minute time limit per run. Experiments were conducted on Amazon EC2 R5 instances (8 vCPUs, 64GB RAM), using standard train-test splits

from OpenML tasks. Meta-features were extracted using pymfe (Alcobaça et al., 2020), based on the descriptors by Rivolli et al. (2022). Additionally, average pipeline performance (score or rank) across training tasks was used as an extra meta-feature.

Meta-models were trained using 10-fold cross-validation on tasks and evaluated across 10 random seeds. Performance was assessed using both standard metrics and ranking-specific ones.

**Performance Metrics.** We report two types of metrics:

- **Ranking metrics**: NDCG and MRR at cutoff positions $k \in \{1, 5, 10\}$ (or $k \in \{1, 10, 100\}$ for AMLB scenarios).

- **System-level metrics**: SCORE (average of the objective metric), TTB (time to find the best pipeline), and AVG_RANK (relative rank position between score- and ranking-based variants).

NDCG measures top-$k$ ranking quality with diminishing weights, while MRR emphasizes early hits by computing the inverse rank of the best item. SCORE reflects general performance, TTB measures efficiency, and AVG_RANK compares method quality relative to others.

## 4  Results

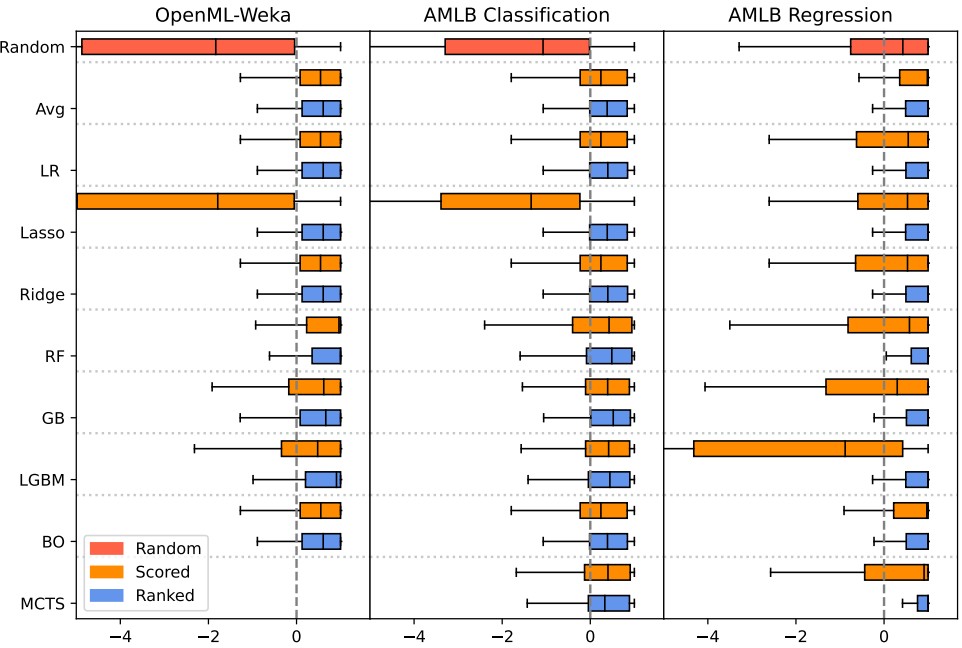

Figure 1: Boxplots of variants (Rank-based in blue and Score-based in orange) performance (x-axis) across tasks for each predictor (y-axis) after scaling the performance values from the mean (0) to best value observed.

Figure 1 shows the performance distribution of Rank- and Score-based variants across tasks. The models (e.g., MCTS, BO, LGBM, RF, Ridge, LR, etc.) are applied to the different datasets using two variants score-based (Scored) and ranking-based (Ranked). Both variants outperform random baselines, but performance is strongly influenced by the predictor type. Rank-based variants typically show tighter, more consistent distributions, suggesting greater robustness across tasks. This is especially evident in AMLB Regression, where Rank-based variants clearly outperform their Score-based counterparts.

Specific models, such as LGBM and RF, exhibit tighter and more favorable performance distributions for both variants, suggesting these models are robust across different tasks. However, simpler models like LR and Avg show narrower performance ranges in Score-based variants. This advantage tends to disappear when the target is transformed to a Rank-based variant. This is clearly visible in the AMLB Regression, although it can also be seen in OpenML-Weka.

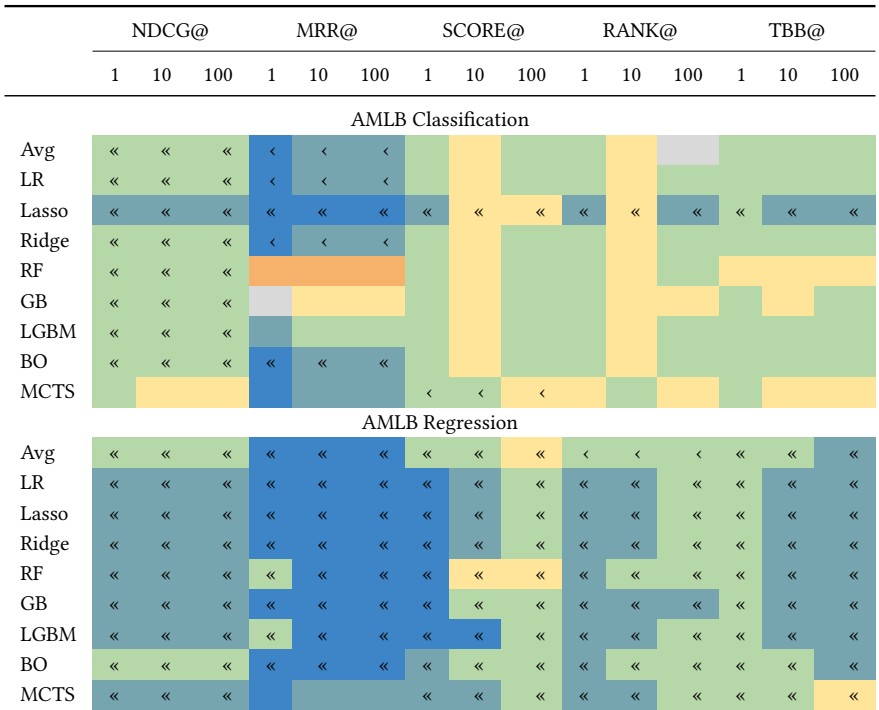

(a) OpenML-Weka

(b) AMLB Classification and Regression

Table 2: Comparison of different approaches evaluated across three datasets: OpenML-Weka, AMLB Classification, and AMLB Regression. Each metric is assessed at three positions ("a", "b", and "c"), measuring performance in terms of NDCG, MRR, SCORE, RANK, and TTB. For the AMLB datasets, the positions are 1, 10, and 100, while for the OpenML-Weka dataset, the positions are 1, 5, and 10. Cell color indicates where the Rank-based approach improves over the Score-based approach (green > 0, dark green > 10%, and blue > 50% of improvements), decreases (yellow < 0, orange < 10%, and red < 50% of improvements), or shows no changes (grey ). The symbols "‹" and "«" indicate statistical significance and "–" indicates not evaluated.

**Rank-based approaches consistently outperform Score-based ones**. Table 2 presents a detailed comparison of different approaches evaluated across the three datasets: OpenML-Weka, AMLB Classification, and AMLB Regression. The performance of these approaches is measured using five metrics (i.e., NDCG, MRR, SCORE, RANK, and TTB) across three cutoff points specific to each dataset (1, 5, and 10 for OpenML-Weka, and 1, 10, and 100 for the AMLB datasets). The table also uses color coding to indicate where the Rank-based approach improves over the Score-based approach. The symbols "‹" and "«" indicate statistical significance using the Wilcoxon signed-rank test (p-value < 0.05) and after Bonferroni adjustment (p-value < 0.005), respectively. Cells without symbols indicate that the observed differences are not statistically significant. The missing entries ("-") for OpenML-Weka are due to the dataset's lack of temporal information, which makes it impossible to calculate TTB, and its absence of information about the components that form the pipelines, preventing the execution of MCTS, as MCTS requires knowledge of pipeline components to structure exploration as a decision tree.

Rank-based approaches consistently outperform Score-based ones across most metrics (green and blue cells). In particular for AMLB Regression, in terms of the SCORE, MRR, and RANK metrics, significant improvements are also observed, particularly at lower positions (e.g., SCORE@1, MRR@1, and RANK@1). For TTB, however, the improvement appears to increase as the number of positions considered increases. This indicates that the evaluated methods are time-effective while maintaining performance even as the cutoff position increases.

Although occasional decreases in Score-based variants are observed (yellow and orange cells), they are generally minor (mostly <10%), with few exceptions. In AMLB Classification, some metrics (e.g., SCORE@b, RANK@b) show smaller losses, yet statistical significance confirms the advantage of Rank-based methods in the majority of cases. A detail of the disaggregated experiments can be seen in Appendix C.

The presence of statistical significance across many of the metrics highlights that the observed improvements are not just due to chance but are statistically significant, especially in datasets OpenML-Weka and AMLB Regression, where the majority of improvements across different datasets and positions are statistically significant. A summary of all experiments corrected using the Bonferroni method can be found in Appendix C.5.

**Ranking metrics correlates with System-level metrics**. Figure 2 shows Spearman correlations between top-position metrics (e.g., NDCG@1, MRR@1, SCORE@1, TTB@1, RANK@1). NDCG and SCORE show strong alignment (e.g., 0.95 in AMLB Classification), confirming that improvements in ranking translate to better task performance. TTB correlates negatively with MRR and NDCG, indicating faster identification of top solutions in Rank-based approaches. RANK also correlates negatively with these metrics, reinforcing its validity as a proxy for ranking quality. Slight variations in OpenML-Weka may be due to its smaller pipeline space. Extended and disaggregated correlation results can be seen in Appendix C.4.

**No significant differences with pairwise models.**. We additionally evaluated AutoFolio on the ASLib scenario (OpenML-Weka-2017, Appendix C.1). As it only returns the top pipeline per task, only MRR@1, SCORE@1, and AVG_RANK@1 could be computed. The Score-based variant slightly outperforms in MRR and AVG_RANK, while Rank-based performs better in SCORE. However, differences were not statistically significant (Wilcoxon p-values > 0.2). Given that AutoFolio relies on pairwise comparisons, which implicitly encode ranking information, this result is expected. Nevertheless, our findings show that it is possible to achieve competitive performance with simpler and less computationally expensive approaches.

**Robustness to Overfitting**. Figure 3 illustrates how increasing Gaussian noise ($\sigma$) affects global ranking quality, as measured by Kendall's Tau, for two datasets: OpenML-Weka (left) and AMLB Classification (right). Each line corresponds to a fixed proportion of meta-training tasks (25%,

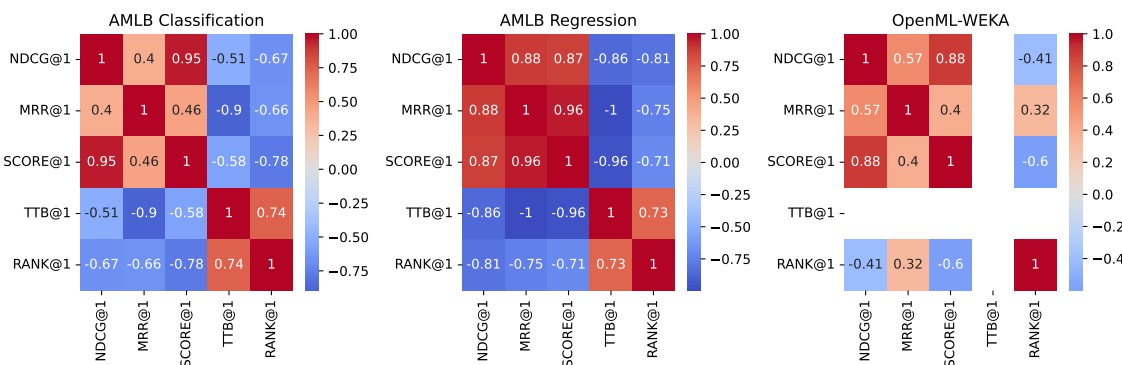

Figure 2: Spearman correlation between metrics at first position.

50%, 75%, 100%), with darker tones indicating more tasks. We compare score-based (orange) and rank-based (blue) averaging.

As noise increases, Kendall's Tau declines, reflecting reduced ranking quality. Rank-based models consistently achieve higher Tau values across all conditions, indicating better preservation of global ordering.

These results provide evidence that rank-based supervision more effectively preserves ordering information, particularly in scenarios where meta-data is limited or corrupted by noise introduced by overfitting or measurement variability. A complementary synthetic experiment can be found in Appendix E

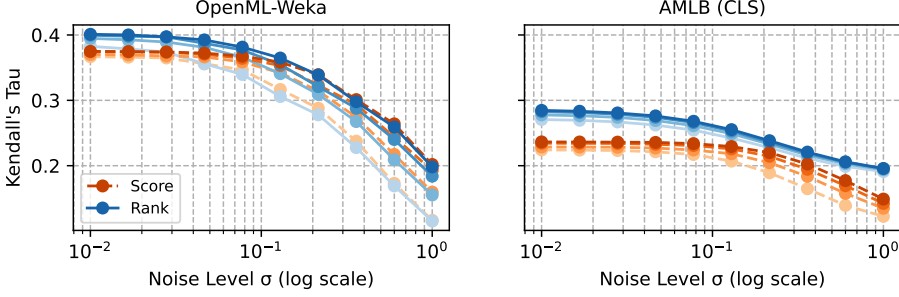

Figure 3: Comparison of Kendall's Tau under increasing Gaussian noise ($\sigma \in [0.01, 1.0]$) in the meta-learning training data for two datasets: OpenML-Weka (left) and AMLB (CLS) (right). Each line corresponds to a fixed number of meta-training tasks {25%, 50%, 75%, 100%}, with darker tones indicating more tasks. Rank-based models (blue) consistently preserve global ranking structure better than score-based models (orange), especially under high noise and limited data.

## 5 Related Work

Pipeline selection is a key challenge in AutoML, with several approaches developed to navigate the combinatorial space of models, preprocessing steps, and hyperparameters (Hutter et al., 2011; Feurer et al., 2015; Vanschoren, 2019). Vanschoren (2019) distinguish between performance predictors and ranking generators. We build on this by showing that reformulating traditional score-based approaches such as BO and MCTS into ranking-based methods can lead to improved pipeline selection.

Several works have explored ranking or pairwise comparison techniques in algorithm selection. AutoFolio (Lindauer et al., 2015), for instance, uses pairwise classifiers trained on performance scores, incorporating transformations like log (Xu et al., 2008) or z-score normalization (Collautti et al., 2013). Other methods also employ pairwise comparisons, though typically based on absolute (Sun and Pfahringer, 2013; Tornede et al., 2020) or relative (Kostovska et al., 2023) performance scores, rather than explicitly modeling ranking positions.

Preselection of pipeline spaces is another common strategy. AutoGluon (Erickson et al., 2020) combines predictions from fixed pipelines using ensembles, while GramML (Vazquez et al., 2022; Vázquez et al., 2023) learns to explore a predefined search space via MCTS. Similarly, portfolio-based methods such as Probabilistic Matrix Factorization (Fusi et al., 2018) and Oboe (Yang et al., 2019) use collaborative filtering to predict performance from historical data.

Ranking has also been used to address cold-start and meta-learning challenges. Auto-sklearn 2.0 (Feurer et al., 2022) and RankML (Laadan et al., 2019) prioritize promising configurations based on prior performance. MetaTPOT (Laadan et al., 2020), in turn, enhances TPOT (Olson and Moore, 2016) by leveraging ranking-based meta-learning to guide evolutionary search.

The approaches described above recognize the importance of using ranking information, whether to preselect the system's components, order them to expedite the search, or as a form of meta-learning to enhance system efficiency. However, the problem of learning to rank and the importance of using positional information has not been addressed, to our knowledge, until this work. We believe this work presents a novel conceptual framework as a basis for learning to rank in AutoML that can potentially be transferred to many other approaches.

## 6 Conclusions and Discussion

This work presents a simple yet effective reformulation of pipeline selection in Automated Machine Learning (AutoML) as a rank prediction problem. By replacing performance score prediction with rank position estimation, we align model objectives more closely with the decision-making structure of the selection task.

Through a systematic comparison of score-based and rank-based approaches across three large-scale OpenML benchmarks, we show that rank-based regression consistently outperforms traditional methods in ranking quality, robustness to noise, and alignment with AutoML objectives such as time-to-solution and final performance. These findings offer valuable practical guidance for AutoML practitioners and system designers.

Moreover, our analysis highlights the advantages of modeling positional information directly, using simple modifications to existing regression models. This lightweight "ranking trick" provides a general and compatible approach that avoids the complexity of traditional listwise or pairwise learning-to-rank techniques.

While this study focuses on scenarios with predefined pipelines and finite search spaces—typical of many AutoML tools—it lays the groundwork for future investigations into other relative ranking techniques, as well as extensions to dynamic pipeline construction or hyperparameter tuning.

Although limited to OpenML, ASLib, and AMLB benchmarks, our results are broadly applicable and point to a promising direction for building more robust, efficient, and decision-oriented AutoML systems.

## 7 Broader Impact Statement

After careful reflection, the authors have determined that this work presents no notable negative impacts to society or the environment.

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

## A  Schematic representation of the proposed approach

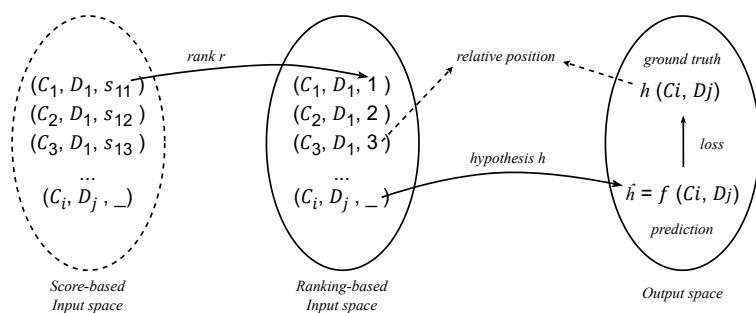

Figure 4: The left block illustrates the original meta-learning data, consisting of triplets $(C_i, D_j, s_{ij})$ where $s_{ij}$ denotes the performance score of configuration $C_i$ on dataset $D_j$. These scores are then transformed into rank ($r$) positions, as shown in the middle block, yielding $(C_i, D_j, h(C_i, D_j))$, where $h$ denotes the rank mapping function. The ranking model $f$ is then trained to predict these positions $\hat{h}$ based on $(C_i, D_j)$ pairs. The loss is computed as the difference between predicted and true ranks. Regardless of the specific loss function used, this change of representation enables the model to directly optimize relative positions rather than absolute performance values, providing increased robustness to score variability, improved generalization, and better alignment with the final selection goal.

## B  OpenML tasks

This appendix provides additional details on the specific datasets used in our study. OpenML tasks are separated by scenario into OpenML-Weka tasks, AMLB-Classification tasks and AMLB-Regression tasks.

### B.1  OpenML-Weka tasks

The OpenML task IDs for OpenML-Weka 2017 as described in ASLib initiative are:

2097, 2098, 2102, 1701, 1702, 1705, 1710, 1711, 1713, 1714, 1715, 1717, 1719, 1720, 1721, 1722, 1723, 1727, 1728, 1730, 1731, 1735, 1736, 1740, 1742, 1744, 1752, 1757, 1764, 10041, 10045, 10046, 10047, 10050, 10053, 10054, 10055, 10067, 10069, 10070, 10071, 10072, 10074, 10075, 10076, 10077, 10079, 10080, 10082, 10083, 10084, 10085, 10086, 7532, 7535, 7536, 125849, 125850, 125851, 125852, 125853, 125854, 125855, 125857, 125859, 125861, 125865, 125866, 125867, 125868, 125869, 125870, 125871, 125873, 125874, 125875, 125876, 125877, 125878, 125879, 125880, 125881, 125884, 125885, 125886, 125887, 125888, 125889, 125890, 125891, 125892, 125894, 125897, 125898, 125899, 125901, 125902, 125905, 125906, 125909, 125910, 125911, 125913, 125914, 125915

We excluded from the selectable datasets those from the AutoML Benchmark (AMLB) that also met the criteria, as they are used in the other experiment. The excluded dataset IDs are as follows: 146818, 359955, and 190146.

### B.2  AMLB-Classification tasks

The selected AMLB IDs for classification tasks are:

2073, 3945, 7593, 10090, 146818, 146820, 167120, 168350, 168757, 168784, 168868, 168909, 168910, 168911, 189354, 189355, 189356, 189922, 190137, 190146, 190392, 190410, 190411, 190412, 211979, 211986, 359953, 359954, 359955, 359956, 359957, 359958, 359959, 359960, 359961, 359962, 359963, 359964, 359965, 359966, 359967, 359968, 359969, 359970, 359971, 359972, 359973, 359974, 359975, 359976, 359977, 359979, 359980, 359981, 359982, 359983, 359984, 359985, 359986, 359987, 359988, 359989, 359990, 359991, 359992, 359993, 359994, 360112, 360113, 360114, 360975

## B.3 AMLB-Regression tasks

The selected AMLB IDs for regression tasks are:

167210, 233211, 233212, 233213, 233214, 233215, 317614, 359929, 359930, 359932, 359933, 359934, 359935, 359936, 359937, 359938, 359939, 359940, 359941, 359942, 359943, 359944, 359945, 359946, 359948, 359949, 359950, 359951, 359952, 360932, 360933, 360945

Task 359931 was excluded due to errors.

## C Detailed experiments results

In this section we break down the results presented in Section 4, separating them into scenarios OpenML-Weka, AMLB-Classification and AMLB-Regression problems.

### C.1 OpenML-Weka

Table 3: Comparison of approaches using ranking metrics NDCG and MRR on ASLib tasks. The best results in each group are highlighted in bold.

| | var | NDCG@1 | NDCG@5 | NDCG@10 | MRR@1 | MRR@5 | MRR@10 |
|---|---|---|---|---|---|---|---|
| Random | | $0.600^{(0.27)}$ | $0.638^{(0.16)}$ | $0.671^{(0.13)}$ | $0.068^{(0.25)}$ | $0.123^{(0.27)}$ | $0.146^{(0.26)}$ |
| Avg | Score | $0.872^{(0.10)}$ | $0.863^{(0.08)}$ | $0.858^{(0.07)}$ | $0.128^{(0.33)}$ | $0.301^{(0.34)}$ | $0.318^{(0.32)}$ |
| | Rank | $0.879^{(0.09)}$ | $0.865^{(0.08)}$ | $0.860^{(0.06)}$ | $0.133^{(0.34)}$ | $0.304^{(0.34)}$ | $0.330^{(0.32)}$ |
| LR | Score | $0.872^{(0.10)}$ | $0.863^{(0.08)}$ | $0.858^{(0.07)}$ | $0.128^{(0.33)}$ | $0.301^{(0.34)}$ | $0.318^{(0.32)}$ |
| | Rank | $0.879^{(0.09)}$ | $0.865^{(0.08)}$ | $0.860^{(0.06)}$ | $0.133^{(0.34)}$ | $0.304^{(0.34)}$ | $0.330^{(0.32)}$ |
| Lasso | Score | $0.610^{(0.26)}$ | $0.638^{(0.16)}$ | $0.668^{(0.13)}$ | $0.076^{(0.27)}$ | $0.144^{(0.29)}$ | $0.165^{(0.28)}$ |
| | Rank | $0.879^{(0.09)}$ | $0.865^{(0.08)}$ | $0.860^{(0.06)}$ | $0.133^{(0.34)}$ | $0.304^{(0.34)}$ | $0.330^{(0.32)}$ |
| Ridge | Score | $0.872^{(0.10)}$ | $0.863^{(0.08)}$ | $0.858^{(0.07)}$ | $0.128^{(0.33)}$ | $0.301^{(0.34)}$ | $0.318^{(0.32)}$ |
| | Rank | $0.879^{(0.09)}$ | $0.865^{(0.08)}$ | $0.860^{(0.06)}$ | $0.133^{(0.34)}$ | $0.304^{(0.34)}$ | $0.330^{(0.32)}$ |
| RF | Score | $0.884^{(0.14)}$ | $0.878^{(0.09)}$ | $0.881^{(0.07)}$ | $0.253^{(0.43)}$ | $0.383^{(0.40)}$ | $0.403^{(0.38)}$ |
| | Rank | $0.899^{(0.13)}$ | $0.892^{(0.08)}$ | $0.898^{(0.06)}$ | $0.296^{(0.46)}$ | $0.426^{(0.41)}$ | $0.448^{(0.39)}$ |
| GB | Score | $0.846^{(0.16)}$ | $0.853^{(0.10)}$ | $0.854^{(0.07)}$ | $0.201^{(0.40)}$ | $0.333^{(0.38)}$ | $0.352^{(0.37)}$ |
| | Rank | $0.872^{(0.12)}$ | $0.867^{(0.08)}$ | $0.866^{(0.06)}$ | $0.156^{(0.36)}$ | $0.321^{(0.35)}$ | $0.344^{(0.34)}$ |
| LGBM | Score | $0.830^{(0.17)}$ | $0.850^{(0.10)}$ | $0.865^{(0.08)}$ | $0.177^{(0.38)}$ | $0.306^{(0.37)}$ | $0.333^{(0.35)}$ |
| | Rank | $0.889^{(0.12)}$ | $0.888^{(0.07)}$ | $0.894^{(0.06)}$ | $0.221^{(0.41)}$ | $0.383^{(0.38)}$ | $0.402^{(0.36)}$ |
| BO | Score | $0.872^{(0.10)}$ | $0.863^{(0.08)}$ | $0.858^{(0.07)}$ | $0.128^{(0.33)}$ | $0.301^{(0.34)}$ | $0.318^{(0.32)}$ |
| | Rank | $0.879^{(0.09)}$ | $0.865^{(0.08)}$ | $0.860^{(0.06)}$ | $0.133^{(0.34)}$ | $0.304^{(0.34)}$ | $0.330^{(0.32)}$ |
| AutoFolio | Score | - | - | - | $0.388^{(0.32)}$ | - | - |
| | Rank | - | - | - | $0.380^{(0.32)}$ | - | - |

Table 4: Comparison of approaches using AutoML metrics average SCORE and average RANK on ASLib tasks. The best results in each group are highlighted in bold.

| | var | SCORE@1 | SCORE@5 | SCORE@10 | RANK@1 | RANK@5 | RANK@10 |
|---|---|---|---|---|---|---|---|
| Random | | $0.765^{(0.23)}$ | $0.856^{(0.17)}$ | $0.866^{(0.17)}$ | - | - | - |
| Avg | Score | $0.854^{(0.18)}$ | $0.869^{(0.17)}$ | $0.873^{(0.17)}$ | $1.041^{(0.20)}$ | $1.006^{(0.08)}$ | $1.107^{(0.31)}$ |
| | Rank | $0.856^{(0.18)}$ | $0.869^{(0.17)}$ | $0.874^{(0.17)}$ | $1.014^{(0.12)}$ | $1.002^{(0.04)}$ | $1.025^{(0.16)}$ |
| LR | Score | $0.854^{(0.18)}$ | $0.869^{(0.17)}$ | $0.873^{(0.17)}$ | $1.041^{(0.20)}$ | $1.006^{(0.08)}$ | $1.109^{(0.31)}$ |
| | Rank | $0.856^{(0.18)}$ | $0.869^{(0.17)}$ | $0.874^{(0.17)}$ | $1.014^{(0.12)}$ | $1.002^{(0.04)}$ | $1.027^{(0.16)}$ |
| Lasso | Score | $0.767^{(0.24)}$ | $0.854^{(0.18)}$ | $0.865^{(0.17)}$ | $1.784^{(0.41)}$ | $1.632^{(0.48)}$ | $1.524^{(0.50)}$ |
| | Rank | $0.856^{(0.18)}$ | $0.869^{(0.17)}$ | $0.874^{(0.17)}$ | $1.146^{(0.35)}$ | $1.137^{(0.34)}$ | $1.103^{(0.30)}$ |
| Ridge | Score | $0.854^{(0.18)}$ | $0.869^{(0.17)}$ | $0.873^{(0.17)}$ | $1.041^{(0.20)}$ | $1.006^{(0.08)}$ | $1.109^{(0.31)}$ |
| | Rank | $0.856^{(0.18)}$ | $0.869^{(0.17)}$ | $0.874^{(0.17)}$ | $1.014^{(0.12)}$ | $1.002^{(0.04)}$ | $1.027^{(0.16)}$ |
| RF | Score | $0.857^{(0.17)}$ | $0.870^{(0.17)}$ | $0.874^{(0.17)}$ | $1.249^{(0.43)}$ | $1.150^{(0.36)}$ | $1.110^{(0.31)}$ |
| | Rank | $0.855^{(0.18)}$ | $0.873^{(0.17)}$ | $0.874^{(0.17)}$ | $1.174^{(0.38)}$ | $1.069^{(0.25)}$ | $1.054^{(0.23)}$ |
| GB | Score | $0.852^{(0.18)}$ | $0.869^{(0.17)}$ | $0.872^{(0.17)}$ | $1.272^{(0.45)}$ | $1.089^{(0.28)}$ | $1.117^{(0.32)}$ |
| | Rank | $0.853^{(0.18)}$ | $0.871^{(0.17)}$ | $0.874^{(0.17)}$ | $1.240^{(0.43)}$ | $1.043^{(0.20)}$ | $1.034^{(0.18)}$ |
| LGBM | Score | $0.850^{(0.18)}$ | $0.870^{(0.17)}$ | $0.874^{(0.17)}$ | $1.416^{(0.49)}$ | $1.237^{(0.43)}$ | $1.131^{(0.34)}$ |
| | Rank | $0.855^{(0.18)}$ | $0.873^{(0.17)}$ | $0.875^{(0.17)}$ | $1.214^{(0.41)}$ | $1.081^{(0.27)}$ | $1.061^{(0.24)}$ |
| BO | Score | $0.854^{(0.18)}$ | $0.869^{(0.17)}$ | $0.873^{(0.17)}$ | $1.039^{(0.19)}$ | $1.006^{(0.08)}$ | $1.109^{(0.31)}$ |
| | Rank | $0.856^{(0.18)}$ | $0.869^{(0.17)}$ | $0.874^{(0.17)}$ | $1.016^{(0.13)}$ | $1.002^{(0.04)}$ | $1.026^{(0.16)}$ |
| AutoFolio | Score | $0.852^{(0.18)}$ | - | - | $1.146^{(0.35)}$ | - | - |
| | Rank | $0.853^{(0.18)}$ | - | - | $1.152^{(0.36)}$ | - | - |

Table 5: Improvement Percentage: Rank-based over Score-based approaches

| | Avg | LR | Lasso | Ridge | RF | GB | LGBM | BO | AutoFolio |
|---|---|---|---|---|---|---|---|---|---|
| NDCG@1 | 0.008 | 0.008 | 0.269 | 0.008 | 0.016 | 0.025 | 0.059 | 0.007 | |
| NDCG@5 | 0.002 | 0.002 | 0.227 | 0.002 | 0.013 | 0.014 | 0.038 | 0.002 | |
| NDCG@10 | 0.003 | 0.003 | 0.192 | 0.003 | 0.017 | 0.011 | 0.030 | 0.002 | |
| MRR@1 | 0.006 | 0.006 | 0.057 | 0.006 | 0.043 | -0.045 | 0.044 | 0.006 | -0.021 |
| MRR@5 | 0.003 | 0.003 | 0.160 | 0.003 | 0.044 | -0.012 | 0.076 | 0.003 | |
| MRR@10 | 0.012 | 0.012 | 0.166 | 0.012 | 0.045 | -0.008 | 0.069 | 0.012 | |
| SCORE@1 | 0.002 | 0.002 | 0.089 | 0.002 | -0.002 | 0.001 | 0.005 | 0.002 | 0.001 |
| SCORE@5 | 0.000 | 0.000 | 0.015 | 0.000 | 0.003 | 0.001 | 0.003 | 0.000 | |
| SCORE@10 | 0.001 | 0.001 | 0.008 | 0.001 | 0.000 | 0.001 | 0.001 | 0.001 | |
| RANK@1 | -0.027 | -0.027 | -0.638 | -0.027 | -0.074 | -0.032 | -0.202 | -0.023 | 0.005 |
| RANK@5 | -0.004 | -0.004 | -0.495 | -0.004 | -0.081 | -0.046 | -0.156 | -0.004 | |
| RANK@10 | -0.082 | -0.082 | -0.421 | -0.082 | -0.056 | -0.083 | -0.070 | -0.083 | |

Table 6: Results of the Wilcoxon signed rank test comparing the rankings induced by the ranked-based vs scored-based strategies. The statistically significant differences ($p\_value < 0.05$) in each group are highlighted in blue.

| | Avg | LR | Lasso | Ridge | RF | GB | LGBM | BO | AutoFolio |
|---|---|---|---|---|---|---|---|---|---|
| NDCG@1 | 0.000 | 0.000 | 0.000 | 0.000 | 0.000 | 0.000 | 0.000 | 0.000 | |
| NDCG@5 | 0.000 | 0.000 | 0.000 | 0.000 | 0.000 | 0.000 | 0.000 | 0.000 | |
| NDCG@10 | 0.000 | 0.000 | 0.000 | 0.000 | 0.000 | 0.000 | 0.000 | 0.000 | |
| MRR@1 | 0.042 | 0.042 | 0.000 | 0.042 | 0.000 | 1.000 | 0.001 | 0.042 | 0.879 |
| MRR@5 | 0.083 | 0.114 | 0.000 | 0.083 | 0.000 | 0.945 | 0.000 | 0.083 | |
| MRR@10 | 0.000 | 0.000 | 0.000 | 0.000 | 0.000 | 0.267 | 0.000 | 0.000 | |
| SCORE@1 | 0.000 | 0.000 | 0.000 | 0.000 | 0.000 | 0.004 | 0.000 | 0.000 | 0.248 |
| SCORE@5 | 0.046 | 0.046 | 0.000 | 0.046 | 0.000 | 0.000 | 0.000 | 0.046 | |
| SCORE@10 | 0.000 | 0.000 | 0.000 | 0.000 | 0.000 | 0.000 | 0.000 | 0.000 | |
| RANK@1 | 0.000 | 0.000 | 0.000 | 0.000 | 0.000 | 0.071 | 0.000 | 0.001 | 0.653 |
| RANK@5 | 0.079 | 0.079 | 0.000 | 0.079 | 0.000 | 0.000 | 0.000 | 0.079 | |
| RANK@10 | 0.000 | 0.000 | 0.000 | 0.000 | 0.000 | 0.000 | 0.000 | 0.000 | |

## C.2 AMLB-Classification

Table 7: Comparison of approaches using ranking metrics NDCG and MRR on AMLB clasification tasks. The best results in each group are highlighted in bold.

| | var | NDCG@1 | NDCG@10 | NDCG@100 | MRR@1 | MRR@10 | MRR@100 |
|---|---|---|---|---|---|---|---|
| Random | | $0.679^{(0.23)}$ | $0.669^{(0.14)}$ | $0.686^{(0.12)}$ | $0.003^{(0.05)}$ | $0.009^{(0.06)}$ | $0.013^{(0.07)}$ |
| Avg | Score | $0.850^{(0.16)}$ | $0.859^{(0.13)}$ | $0.872^{(0.09)}$ | $0.001^{(0.04)}$ | $0.028^{(0.09)}$ | $0.035^{(0.09)}$ |
| | Rank | $0.884^{(0.12)}$ | $0.890^{(0.10)}$ | $0.878^{(0.08)}$ | $0.023^{(0.15)}$ | $0.040^{(0.16)}$ | $0.049^{(0.16)}$ |
| LR | Score | $0.850^{(0.16)}$ | $0.860^{(0.13)}$ | $0.872^{(0.09)}$ | $0.001^{(0.04)}$ | $0.028^{(0.09)}$ | $0.035^{(0.09)}$ |
| | Rank | $0.885^{(0.12)}$ | $0.890^{(0.10)}$ | $0.878^{(0.08)}$ | $0.023^{(0.15)}$ | $0.041^{(0.17)}$ | $0.049^{(0.16)}$ |
| Lasso | Score | $0.653^{(0.23)}$ | $0.662^{(0.14)}$ | $0.684^{(0.12)}$ | $0.003^{(0.05)}$ | $0.009^{(0.07)}$ | $0.013^{(0.07)}$ |
| | Rank | $0.884^{(0.12)}$ | $0.890^{(0.10)}$ | $0.878^{(0.08)}$ | $0.023^{(0.15)}$ | $0.039^{(0.16)}$ | $0.048^{(0.16)}$ |
| Ridge | Score | $0.850^{(0.16)}$ | $0.860^{(0.13)}$ | $0.872^{(0.09)}$ | $0.001^{(0.04)}$ | $0.028^{(0.09)}$ | $0.035^{(0.09)}$ |
| | Rank | $0.885^{(0.12)}$ | $0.890^{(0.10)}$ | $0.878^{(0.08)}$ | $0.023^{(0.15)}$ | $0.041^{(0.17)}$ | $0.049^{(0.16)}$ |
| RF | Score | $0.846^{(0.18)}$ | $0.845^{(0.14)}$ | $0.847^{(0.11)}$ | $0.044^{(0.20)}$ | $0.067^{(0.22)}$ | $0.075^{(0.22)}$ |
| | Rank | $0.878^{(0.14)}$ | $0.876^{(0.11)}$ | $0.870^{(0.09)}$ | $0.030^{(0.17)}$ | $0.047^{(0.18)}$ | $0.056^{(0.18)}$ |
| GB | Score | $0.870^{(0.14)}$ | $0.872^{(0.11)}$ | $0.872^{(0.09)}$ | $0.023^{(0.15)}$ | $0.044^{(0.16)}$ | $0.051^{(0.16)}$ |
| | Rank | $0.893^{(0.12)}$ | $0.894^{(0.09)}$ | $0.878^{(0.07)}$ | $0.023^{(0.15)}$ | $0.041^{(0.17)}$ | $0.050^{(0.17)}$ |
| LGBM | Score | $0.868^{(0.15)}$ | $0.870^{(0.11)}$ | $0.867^{(0.09)}$ | $0.015^{(0.12)}$ | $0.038^{(0.15)}$ | $0.045^{(0.15)}$ |
| | Rank | $0.888^{(0.12)}$ | $0.885^{(0.09)}$ | $0.874^{(0.08)}$ | $0.021^{(0.14)}$ | $0.039^{(0.16)}$ | $0.047^{(0.16)}$ |
| BO | Score | $0.850^{(0.16)}$ | $0.860^{(0.13)}$ | $0.872^{(0.09)}$ | $0.001^{(0.04)}$ | $0.028^{(0.09)}$ | $0.035^{(0.09)}$ |
| | Rank | $0.884^{(0.12)}$ | $0.890^{(0.10)}$ | $0.878^{(0.08)}$ | $0.023^{(0.15)}$ | $0.041^{(0.17)}$ | $0.049^{(0.16)}$ |
| MCTS | Score | $0.864^{(0.15)}$ | $0.855^{(0.13)}$ | $0.837^{(0.10)}$ | $0.030^{(0.17)}$ | $0.052^{(0.18)}$ | $0.061^{(0.18)}$ |
| | Rank | $0.876^{(0.13)}$ | $0.852^{(0.10)}$ | $0.824^{(0.08)}$ | $0.051^{(0.22)}$ | $0.059^{(0.22)}$ | $0.068^{(0.22)}$ |

Table 8: Comparison of approaches using AutoML metrics average SCORE and average RANK on AMLB classification tasks. The best results in each group are highlighted in bold.

| | var | SCORE@1 | SCORE@10 | SCORE@100 | RANK@1 | RANK@10 | RANK@100 |
|---|---|---|---|---|---|---|---|
| Random | | $0.558^{(0.22)}$ | $0.723^{(0.18)}$ | $0.777^{(0.17)}$ | - | - | - |
| Avg | Score | $0.675^{(0.20)}$ | $0.723^{(0.19)}$ | $0.767^{(0.17)}$ | $1.500^{(0.50)}$ | $1.238^{(0.43)}$ | $1.172^{(0.38)}$ |
| | Rank | $0.701^{(0.18)}$ | $0.720^{(0.18)}$ | $0.772^{(0.17)}$ | $1.366^{(0.48)}$ | $1.399^{(0.49)}$ | $1.172^{(0.38)}$ |
| LR | Score | $0.675^{(0.20)}$ | $0.723^{(0.19)}$ | $0.767^{(0.17)}$ | $1.504^{(0.50)}$ | $1.238^{(0.43)}$ | $1.170^{(0.38)}$ |
| | Rank | $0.702^{(0.18)}$ | $0.720^{(0.18)}$ | $0.772^{(0.17)}$ | $1.361^{(0.48)}$ | $1.392^{(0.49)}$ | $1.168^{(0.37)}$ |
| Lasso | Score | $0.545^{(0.22)}$ | $0.724^{(0.18)}$ | $0.776^{(0.17)}$ | $1.813^{(0.39)}$ | $1.469^{(0.50)}$ | $1.510^{(0.50)}$ |
| | Rank | $0.701^{(0.18)}$ | $0.720^{(0.18)}$ | $0.772^{(0.17)}$ | $1.156^{(0.36)}$ | $1.473^{(0.50)}$ | $1.331^{(0.47)}$ |
| Ridge | Score | $0.675^{(0.20)}$ | $0.723^{(0.19)}$ | $0.767^{(0.17)}$ | $1.504^{(0.50)}$ | $1.238^{(0.43)}$ | $1.170^{(0.38)}$ |
| | Rank | $0.702^{(0.18)}$ | $0.720^{(0.18)}$ | $0.772^{(0.17)}$ | $1.361^{(0.48)}$ | $1.392^{(0.49)}$ | $1.168^{(0.37)}$ |
| RF | Score | $0.674^{(0.19)}$ | $0.739^{(0.17)}$ | $0.779^{(0.17)}$ | $1.485^{(0.50)}$ | $1.361^{(0.48)}$ | $1.306^{(0.46)}$ |
| | Rank | $0.688^{(0.19)}$ | $0.737^{(0.18)}$ | $0.781^{(0.17)}$ | $1.427^{(0.49)}$ | $1.437^{(0.50)}$ | $1.272^{(0.44)}$ |
| GB | Score | $0.684^{(0.20)}$ | $0.734^{(0.18)}$ | $0.768^{(0.17)}$ | $1.459^{(0.50)}$ | $1.282^{(0.45)}$ | $1.193^{(0.39)}$ |
| | Rank | $0.702^{(0.18)}$ | $0.731^{(0.18)}$ | $0.772^{(0.17)}$ | $1.375^{(0.48)}$ | $1.377^{(0.48)}$ | $1.199^{(0.40)}$ |
| LGBM | Score | $0.688^{(0.19)}$ | $0.735^{(0.18)}$ | $0.770^{(0.17)}$ | $1.423^{(0.49)}$ | $1.327^{(0.47)}$ | $1.225^{(0.42)}$ |
| | Rank | $0.697^{(0.18)}$ | $0.734^{(0.18)}$ | $0.773^{(0.17)}$ | $1.420^{(0.49)}$ | $1.377^{(0.48)}$ | $1.208^{(0.41)}$ |
| BO | Score | $0.675^{(0.20)}$ | $0.723^{(0.19)}$ | $0.767^{(0.17)}$ | $1.503^{(0.50)}$ | $1.238^{(0.43)}$ | $1.170^{(0.38)}$ |
| | Rank | $0.701^{(0.18)}$ | $0.720^{(0.18)}$ | $0.772^{(0.17)}$ | $1.358^{(0.48)}$ | $1.392^{(0.49)}$ | $1.168^{(0.37)}$ |
| MCTS | Score | $0.686^{(0.19)}$ | $0.725^{(0.18)}$ | $0.774^{(0.17)}$ | $1.306^{(0.46)}$ | $1.454^{(0.50)}$ | $1.286^{(0.45)}$ |
| | Rank | $0.698^{(0.18)}$ | $0.733^{(0.18)}$ | $0.773^{(0.17)}$ | $1.325^{(0.47)}$ | $1.368^{(0.48)}$ | $1.341^{(0.47)}$ |

Table 9: Comparison of approaches using average TTB on AMLB classification tasks. The best results in each group are highlighted in bold.

| | var | TTB@1 | TTB@10 | TTB@100 |
|---|---|---|---|---|
| Random | | $5041.482^{(3583.41)}$ | $4908.858^{(3639.28)}$ | $4081.935^{(3736.80)}$ |
| Avg | Score | $5043.765^{(3580.72)}$ | $4448.601^{(3677.92)}$ | $3033.816^{(3525.23)}$ |
| | Rank | $4874.958^{(3624.69)}$ | $4362.415^{(3660.09)}$ | $2926.108^{(3514.38)}$ |
| LR | Score | $5043.765^{(3580.72)}$ | $4437.394^{(3679.17)}$ | $3033.332^{(3524.90)}$ |
| | Rank | $4874.908^{(3624.76)}$ | $4352.566^{(3662.46)}$ | $2924.392^{(3515.49)}$ |
| Lasso | Score | $5032.579^{(3583.86)}$ | $4876.417^{(3636.97)}$ | $4203.129^{(3747.74)}$ |
| | Rank | $4874.958^{(3624.69)}$ | $4362.433^{(3660.07)}$ | $2925.991^{(3514.42)}$ |
| Ridge | Score | $5043.765^{(3580.72)}$ | $4437.394^{(3679.17)}$ | $3033.332^{(3524.90)}$ |
| | Rank | $4874.908^{(3624.76)}$ | $4352.566^{(3662.46)}$ | $2924.392^{(3515.49)}$ |
| RF | Score | $4787.428^{(3668.17)}$ | $4249.099^{(3724.16)}$ | $2849.469^{(3502.55)}$ |
| | Rank | $4934.319^{(3615.37)}$ | $4578.992^{(3718.60)}$ | $2939.404^{(3455.19)}$ |
| GB | Score | $4927.551^{(3616.19)}$ | $4347.207^{(3679.70)}$ | $3061.660^{(3518.02)}$ |
| | Rank | $4873.310^{(3623.17)}$ | $4367.823^{(3672.12)}$ | $2988.906^{(3558.88)}$ |
| LGBM | Score | $4947.902^{(3602.84)}$ | $4509.409^{(3674.22)}$ | $3138.629^{(3504.85)}$ |
| | Rank | $4878.782^{(3613.86)}$ | $4369.698^{(3668.37)}$ | $3075.300^{(3562.53)}$ |
| BO | Score | $5043.765^{(3580.72)}$ | $4437.394^{(3679.17)}$ | $3033.415^{(3524.88)}$ |
| | Rank | $4874.908^{(3624.76)}$ | $4352.566^{(3662.46)}$ | $2924.351^{(3515.50)}$ |
| MCTS | Score | $4812.553^{(3628.41)}$ | $4218.461^{(3735.58)}$ | $2965.364^{(3726.11)}$ |
| | Rank | $4631.098^{(3651.16)}$ | $4330.997^{(3729.38)}$ | $3165.093^{(3672.53)}$ |

Table 10: Improvement Percentage: Rank-based over Score-based approaches

|  | Avg | LR | Lasso | Ridge | RF | GB | LGBM | BO | MCTS |
|---|---|---|---|---|---|---|---|---|---|
| NDCG@1 | 0.040 | 0.041 | 0.353 | 0.041 | 0.038 | 0.027 | 0.023 | 0.041 | 0.014 |
| NDCG@10 | 0.036 | 0.036 | 0.344 | 0.036 | 0.037 | 0.025 | 0.017 | 0.036 | -0.003 |
| NDCG@100 | 0.007 | 0.007 | 0.284 | 0.007 | 0.027 | 0.007 | 0.008 | 0.007 | -0.016 |
| MRR@1 | 15.000 | 15.000 | 7.000 | 15.000 | -0.323 | 0.000 | 0.364 | 15.000 | 0.714 |
| MRR@10 | 0.430 | 0.457 | 3.446 | 0.457 | -0.297 | -0.061 | 0.032 | 0.457 | 0.145 |
| MRR@100 | 0.374 | 0.398 | 2.641 | 0.398 | -0.260 | -0.027 | 0.036 | 0.398 | 0.110 |
| SCORE@1 | 0.039 | 0.040 | 0.287 | 0.040 | 0.021 | 0.026 | 0.013 | 0.040 | 0.017 |
| SCORE@10 | -0.005 | -0.004 | -0.007 | -0.004 | -0.003 | -0.004 | -0.002 | -0.005 | 0.011 |
| SCORE@100 | 0.007 | 0.007 | -0.005 | 0.007 | 0.003 | 0.006 | 0.004 | 0.007 | -0.001 |
| TTB@1 | -0.033 | -0.033 | -0.031 | -0.033 | 0.031 | -0.011 | -0.014 | -0.033 | -0.038 |
| TTB@10 | -0.019 | -0.019 | -0.105 | -0.019 | 0.078 | 0.005 | -0.031 | -0.019 | 0.027 |
| TTB@100 | -0.036 | -0.036 | -0.304 | -0.036 | 0.032 | -0.024 | -0.020 | -0.036 | 0.067 |
| RANK@1 | -0.089 | -0.096 | -0.362 | -0.096 | -0.039 | -0.058 | -0.002 | -0.097 | 0.015 |
| RANK@10 | 0.130 | 0.124 | 0.003 | 0.124 | 0.056 | 0.075 | 0.038 | 0.124 | -0.059 |
| RANK@100 | 0.000 | -0.002 | -0.118 | -0.002 | -0.026 | 0.005 | -0.014 | -0.002 | 0.043 |

Table 11: Results of the Wilcoxon signed rank test comparing the rankings induced by the ranked-based vs scored-based strategies. The statistically significant differences ($p\_value < 0.05$) in each group are highlighted in blue.

|  | Avg | LR | Lasso | Ridge | RF | GB | LGBM | BO | MCTS |
|---|---|---|---|---|---|---|---|---|---|
| NDCG@1 | 0.000 | 0.000 | 0.000 | 0.000 | 0.001 | 0.001 | 0.008 | 0.000 | 0.137 |
| NDCG@10 | 0.000 | 0.000 | 0.000 | 0.000 | 0.000 | 0.000 | 0.000 | 0.000 | 0.992 |
| NDCG@100 | 0.004 | 0.003 | 0.000 | 0.003 | 0.000 | 0.002 | 0.000 | 0.003 | 1.000 |
| MRR@1 | 0.000 | 0.000 | 0.000 | 0.000 | 0.926 | 0.500 | 0.197 | 0.000 | 0.004 |
| MRR@10 | 0.108 | 0.081 | 0.000 | 0.081 | 0.981 | 0.618 | 0.411 | 0.081 | 0.356 |
| MRR@100 | 0.006 | 0.005 | 0.000 | 0.005 | 0.943 | 0.462 | 0.252 | 0.005 | 0.995 |
| SCORE@1 | 0.000 | 0.000 | 0.000 | 0.000 | 0.007 | 0.002 | 0.107 | 0.000 | 0.359 |
| SCORE@10 | 1.000 | 1.000 | 0.843 | 1.000 | 0.999 | 1.000 | 0.930 | 1.000 | 0.000 |
| SCORE@100 | 0.001 | 0.001 | 0.144 | 0.001 | 0.022 | 0.012 | 0.018 | 0.001 | 0.578 |
| TTB@1 | 0.000 | 0.000 | 0.000 | 0.000 | 0.990 | 0.258 | 0.081 | 0.000 | 0.000 |
| TTB@10 | 0.945 | 0.933 | 0.000 | 0.933 | 1.000 | 0.974 | 0.338 | 0.933 | 0.978 |
| TTB@100 | 0.000 | 0.000 | 0.000 | 0.000 | 0.971 | 0.001 | 0.006 | 0.000 | 0.863 |
| AVG_RANK@1 | 0.000 | 0.000 | 0.000 | 0.000 | 0.053 | 0.007 | 0.467 | 0.000 | 0.746 |
| AVG_RANK@10 | 1.000 | 1.000 | 0.546 | 1.000 | 0.988 | 0.999 | 0.946 | 1.000 | 0.006 |
| AVG_RANK@100 | 0.500 | 0.449 | 0.000 | 0.449 | 0.118 | 0.595 | 0.247 | 0.449 | 0.968 |

## C.3 AMLB-Regression

Table 12: Comparison of approaches using ranking metrics NDCG and MRR on AMLB regression tasks. The best results in each group are highlighted in bold.

| | var | NDCG@1 | NDCG@10 | NDCG@100 | MRR@1 | MRR@10 | MRR@100 |
|---|---|---|---|---|---|---|---|
| Random | | $0.711^{(0.20)}$ | $0.715^{(0.12)}$ | $0.736^{(0.11)}$ | $0.000^{(0.00)}$ | $0.010^{(0.06)}$ | $0.016^{(0.06)}$ |
| Avg | Score | $0.778^{(0.23)}$ | $0.789^{(0.16)}$ | $0.817^{(0.10)}$ | $0.039^{(0.19)}$ | $0.062^{(0.20)}$ | $0.068^{(0.20)}$ |
| | Rank | $0.846^{(0.19)}$ | $0.855^{(0.15)}$ | $0.868^{(0.11)}$ | $0.073^{(0.26)}$ | $0.113^{(0.28)}$ | $0.123^{(0.28)}$ |
| LR | Score | $0.723^{(0.21)}$ | $0.715^{(0.13)}$ | $0.736^{(0.11)}$ | $0.012^{(0.11)}$ | $0.016^{(0.11)}$ | $0.022^{(0.11)}$ |
| | Rank | $0.845^{(0.20)}$ | $0.855^{(0.15)}$ | $0.869^{(0.11)}$ | $0.079^{(0.27)}$ | $0.116^{(0.29)}$ | $0.125^{(0.28)}$ |
| Lasso | Score | $0.721^{(0.22)}$ | $0.714^{(0.13)}$ | $0.736^{(0.11)}$ | $0.012^{(0.11)}$ | $0.016^{(0.11)}$ | $0.022^{(0.11)}$ |
| | Rank | $0.846^{(0.19)}$ | $0.855^{(0.15)}$ | $0.869^{(0.11)}$ | $0.073^{(0.26)}$ | $0.113^{(0.28)}$ | $0.123^{(0.28)}$ |
| Ridge | Score | $0.721^{(0.22)}$ | $0.714^{(0.13)}$ | $0.736^{(0.11)}$ | $0.012^{(0.11)}$ | $0.016^{(0.11)}$ | $0.022^{(0.11)}$ |
| | Rank | $0.845^{(0.20)}$ | $0.855^{(0.15)}$ | $0.869^{(0.11)}$ | $0.079^{(0.27)}$ | $0.116^{(0.29)}$ | $0.125^{(0.28)}$ |
| RF | Score | $0.702^{(0.22)}$ | $0.711^{(0.13)}$ | $0.737^{(0.11)}$ | $0.000^{(0.00)}$ | $0.009^{(0.05)}$ | $0.014^{(0.05)}$ |
| | Rank | $0.858^{(0.17)}$ | $0.858^{(0.15)}$ | $0.863^{(0.13)}$ | $0.030^{(0.17)}$ | $0.062^{(0.19)}$ | $0.075^{(0.19)}$ |
| GB | Score | $0.693^{(0.23)}$ | $0.708^{(0.13)}$ | $0.734^{(0.11)}$ | $0.003^{(0.05)}$ | $0.008^{(0.06)}$ | $0.014^{(0.06)}$ |
| | Rank | $0.835^{(0.20)}$ | $0.847^{(0.16)}$ | $0.868^{(0.11)}$ | $0.067^{(0.25)}$ | $0.093^{(0.26)}$ | $0.105^{(0.26)}$ |
| LGBM | Score | $0.613^{(0.26)}$ | $0.663^{(0.18)}$ | $0.724^{(0.12)}$ | $0.000^{(0.00)}$ | $0.006^{(0.03)}$ | $0.012^{(0.03)}$ |
| | Rank | $0.828^{(0.20)}$ | $0.839^{(0.16)}$ | $0.859^{(0.12)}$ | $0.055^{(0.23)}$ | $0.076^{(0.24)}$ | $0.089^{(0.23)}$ |
| BO | Score | $0.796^{(0.21)}$ | $0.804^{(0.14)}$ | $0.816^{(0.10)}$ | $0.027^{(0.16)}$ | $0.062^{(0.19)}$ | $0.070^{(0.19)}$ |
| | Rank | $0.846^{(0.19)}$ | $0.855^{(0.15)}$ | $0.869^{(0.11)}$ | $0.079^{(0.27)}$ | $0.116^{(0.29)}$ | $0.125^{(0.28)}$ |
| MCTS | Score | $0.733^{(0.23)}$ | $0.732^{(0.19)}$ | $0.754^{(0.14)}$ | $0.052^{(0.22)}$ | $0.066^{(0.22)}$ | $0.072^{(0.22)}$ |
| | Rank | $0.878^{(0.17)}$ | $0.845^{(0.11)}$ | $0.851^{(0.09)}$ | $0.085^{(0.28)}$ | $0.085^{(0.28)}$ | $0.091^{(0.28)}$ |

Table 13: Comparison of approaches using AutoML metrics average SCORE and average RANK on AMLB regression tasks. The best results in each group are highlighted in bold.

| | var | SCORE@1 | SCORE@10 | SCORE@100 | RANK@1 | RANK@10 | RANK@100 |
|---|---|---|---|---|---|---|---|
| Random | | $-1.90\text{e+}22^{(3.02e+23)}$ | $-2.13\text{e+}06^{(1.06e+07)}$ | $-1.94\text{e+}06^{(9.89e+06)}$ | - | - | - |
| Avg | Score | $-2.28\text{e+}06^{(1.07e+07)}$ | $-2.01\text{e+}06^{(1.01e+07)}$ | $-1.90\text{e+}06^{(9.70e+06)}$ | $1.418^{(0.49)}$ | $1.403^{(0.49)}$ | $1.358^{(0.48)}$ |
| | Rank | $-2.10\text{e+}06^{(1.07e+07)}$ | $-1.91\text{e+}06^{(9.70e+06)}$ | $-1.91\text{e+}06^{(9.70e+06)}$ | $1.382^{(0.49)}$ | $1.352^{(0.48)}$ | $1.264^{(0.44)}$ |
| LR | Score | $-1.26\text{e+}14^{(2.27e+15)}$ | $-2.19\text{e+}06^{(1.10e+07)}$ | $-1.95\text{e+}06^{(9.95e+06)}$ | $1.721^{(0.45)}$ | $1.630^{(0.48)}$ | $1.464^{(0.50)}$ |
| | Rank | $-2.08\text{e+}06^{(1.06e+07)}$ | $-1.91\text{e+}06^{(9.70e+06)}$ | $-1.91\text{e+}06^{(9.70e+06)}$ | $1.267^{(0.44)}$ | $1.321^{(0.47)}$ | $1.339^{(0.47)}$ |
| Lasso | Score | $-1.26\text{e+}14^{(2.27e+15)}$ | $-2.19\text{e+}06^{(1.10e+07)}$ | $-1.95\text{e+}06^{(9.94e+06)}$ | $1.721^{(0.45)}$ | $1.627^{(0.48)}$ | $1.464^{(0.50)}$ |
| | Rank | $-2.10\text{e+}06^{(1.07e+07)}$ | $-1.91\text{e+}06^{(9.70e+06)}$ | $-1.91\text{e+}06^{(9.70e+06)}$ | $1.267^{(0.44)}$ | $1.327^{(0.47)}$ | $1.339^{(0.47)}$ |
| Ridge | Score | $-1.26\text{e+}14^{(2.27e+15)}$ | $-2.19\text{e+}06^{(1.10e+07)}$ | $-1.95\text{e+}06^{(9.94e+06)}$ | $1.721^{(0.45)}$ | $1.633^{(0.48)}$ | $1.467^{(0.50)}$ |
| | Rank | $-2.08\text{e+}06^{(1.06e+07)}$ | $-1.91\text{e+}06^{(9.70e+06)}$ | $-1.91\text{e+}06^{(9.70e+06)}$ | $1.267^{(0.44)}$ | $1.324^{(0.47)}$ | $1.339^{(0.47)}$ |
| RF | Score | $-2.19\text{e+}37^{(3.97e+38)}$ | $-2.11\text{e+}06^{(1.06e+07)}$ | $-1.94\text{e+}06^{(9.92e+06)}$ | $1.782^{(0.41)}$ | $1.506^{(0.50)}$ | $1.424^{(0.49)}$ |
| | Rank | $-2.34\text{e+}06^{(1.17e+07)}$ | $-2.23\text{e+}06^{(1.14e+07)}$ | $-1.98\text{e+}06^{(1.01e+07)}$ | $1.215^{(0.41)}$ | $1.467^{(0.50)}$ | $1.379^{(0.49)}$ |
| GB | Score | $-3.90\text{e+}58^{(2.21e+59)}$ | $-2.13\text{e+}06^{(1.07e+07)}$ | $-1.95\text{e+}06^{(9.91e+06)}$ | $1.742^{(0.44)}$ | $1.633^{(0.48)}$ | $1.521^{(0.50)}$ |
| | Rank | $-2.09\text{e+}06^{(1.06e+07)}$ | $-1.97\text{e+}06^{(1.00e+07)}$ | $-1.91\text{e+}06^{(9.72e+06)}$ | $1.242^{(0.43)}$ | $1.309^{(0.46)}$ | $1.279^{(0.45)}$ |
| LGBM | Score | $-6.71\text{e+}57^{(4.93e+58)}$ | $-1.40\text{e+}34^{(1.33e+35)}$ | $-1.93\text{e+}06^{(9.82e+06)}$ | $1.833^{(0.37)}$ | $1.655^{(0.48)}$ | $1.439^{(0.50)}$ |
| | Rank | $-5.53\text{e+}06^{(6.37e+07)}$ | $-1.98\text{e+}06^{(1.01e+07)}$ | $-1.92\text{e+}06^{(9.78e+06)}$ | $1.164^{(0.37)}$ | $1.309^{(0.46)}$ | $1.306^{(0.46)}$ |
| BO | Score | $-2.39\text{e+}06^{(1.17e+07)}$ | $-2.01\text{e+}06^{(1.01e+07)}$ | $-1.91\text{e+}06^{(9.74e+06)}$ | $1.542^{(0.50)}$ | $1.436^{(0.50)}$ | $1.367^{(0.48)}$ |
| | Rank | $-2.08\text{e+}06^{(1.06e+07)}$ | $-1.91\text{e+}06^{(9.70e+06)}$ | $-1.91\text{e+}06^{(9.70e+06)}$ | $1.370^{(0.48)}$ | $1.367^{(0.48)}$ | $1.273^{(0.45)}$ |
| MCTS | Score | $-2.31\text{e+}06^{(1.08e+07)}$ | $-2.13\text{e+}06^{(1.01e+07)}$ | $-1.92\text{e+}06^{(9.70e+06)}$ | $1.639^{(0.48)}$ | $1.597^{(0.49)}$ | $1.379^{(0.49)}$ |
| | Rank | $-1.92\text{e+}06^{(9.70e+06)}$ | $-1.91\text{e+}06^{(9.70e+06)}$ | $-1.91\text{e+}06^{(9.70e+06)}$ | $1.330^{(0.47)}$ | $1.348^{(0.48)}$ | $1.309^{(0.46)}$ |

Table 14: Comparison of approaches using average TTB on AMLB regression tasks. The best results in each group are highlighted in bold.

| | var | TTB@1 | TTB@10 | TTB@100 |
|---|---|---|---|---|
| Random | | 3586.084 $^{(1832.81)}$ | 3453.575 $^{(1911.90)}$ | 2761.787 $^{(2145.84)}$ |
| Avg | Score | 3424.625 $^{(1951.65)}$ | 2903.705 $^{(2142.14)}$ | 2111.220 $^{(1812.69)}$ |
| | Rank | 3290.290 $^{(2042.10)}$ | 2708.923 $^{(2060.88)}$ | 1859.865 $^{(1852.73)}$ |
| LR | Score | 3545.273 $^{(1864.66)}$ | 3455.948 $^{(1928.41)}$ | 2747.323 $^{(2190.03)}$ |
| | Rank | 3255.857 $^{(2045.68)}$ | 2690.243 $^{(2056.09)}$ | 1859.944 $^{(1852.76)}$ |
| Lasso | Score | 3545.273 $^{(1864.66)}$ | 3455.916 $^{(1928.47)}$ | 2729.861 $^{(2181.15)}$ |
| | Rank | 3290.291 $^{(2042.10)}$ | 2708.788 $^{(2061.04)}$ | 1859.427 $^{(1852.95)}$ |
| Ridge | Score | 3545.273 $^{(1864.66)}$ | 3455.916 $^{(1928.47)}$ | 2737.503 $^{(2177.95)}$ |
| | Rank | 3255.857 $^{(2045.68)}$ | 2690.243 $^{(2056.09)}$ | 1859.944 $^{(1852.76)}$ |
| RF | Score | 3586.084 $^{(1832.81)}$ | 3458.212 $^{(1897.20)}$ | 2843.086 $^{(2124.43)}$ |
| | Rank | 3477.630 $^{(1924.12)}$ | 3024.837 $^{(2140.76)}$ | 1941.372 $^{(2036.63)}$ |
| GB | Score | 3572.849 $^{(1842.57)}$ | 3485.569 $^{(1903.98)}$ | 2798.697 $^{(2151.61)}$ |
| | Rank | 3317.121 $^{(2027.53)}$ | 2849.755 $^{(2131.73)}$ | 1731.679 $^{(1866.51)}$ |
| LGBM | Score | 3586.084 $^{(1832.81)}$ | 3416.294 $^{(1896.80)}$ | 2541.332 $^{(2095.30)}$ |
| | Rank | 3371.637 $^{(1979.92)}$ | 2941.128 $^{(2091.28)}$ | 1757.136 $^{(1800.78)}$ |
| BO | Score | 3477.129 $^{(1919.76)}$ | 2986.739 $^{(2124.44)}$ | 2175.204 $^{(1870.60)}$ |
| | Rank | 3255.857 $^{(2045.68)}$ | 2690.243 $^{(2056.08)}$ | 1859.945 $^{(1852.76)}$ |
| MCTS | Score | 3378.009 $^{(1983.92)}$ | 3145.425 $^{(2127.41)}$ | 2319.453 $^{(2258.28)}$ |
| | Rank | 3148.875 $^{(1956.02)}$ | 3137.958 $^{(1963.27)}$ | 2549.071 $^{(2148.62)}$ |

Table 15: Improvement Percentage: Rank-based over Score-based approaches

| | Avg | LR | Lasso | Ridge | RF | GB | LGBM | BO | MCTS |
|---|---|---|---|---|---|---|---|---|---|
| NDCG@1 | 0.088 | 0.169 | 0.173 | 0.173 | 0.221 | 0.204 | 0.351 | 0.063 | 0.198 |
| NDCG@10 | 0.083 | 0.196 | 0.198 | 0.198 | 0.207 | 0.195 | 0.266 | 0.064 | 0.154 |
| NDCG@100 | 0.063 | 0.180 | 0.180 | 0.180 | 0.171 | 0.183 | 0.186 | 0.065 | 0.129 |
| MRR@1 | 0.846 | 5.500 | 5.000 | 5.500 | | 21.000 | | 1.889 | 0.647 |
| MRR@10 | 0.826 | 6.433 | 6.107 | 6.282 | 6.233 | 10.791 | 12.154 | 0.862 | 0.292 |
| MRR@100 | 0.791 | 4.689 | 4.466 | 4.592 | 4.310 | 6.432 | 6.370 | 0.785 | 0.258 |
| SCORE@1 | -8.08e-02 | -1.00e+00 | -1.00e+00 | -1.00e+00 | -1.00e+00 | -1.00e+00 | -1.00e+00 | -1.28e-01 | -1.70e-01 |
| SCORE@10 | -4.60e-02 | -1.27e-01 | -1.27e-01 | -1.27e-01 | 5.83e-02 | -7.56e-02 | -1.00e+00 | -4.69e-02 | -1.01e-01 |
| SCORE@100 | 2.48e-03 | -2.13e-02 | -1.98e-02 | -1.98e-02 | 2.00e-02 | -1.81e-02 | -4.64e-03 | -6.35e-04 | -3.94e-03 |
| TTB@1 | -0.039 | -0.082 | -0.072 | -0.082 | -0.030 | -0.072 | -0.060 | -0.064 | -0.068 |
| TTB@10 | -0.067 | -0.222 | -0.216 | -0.222 | -0.125 | -0.182 | -0.139 | -0.099 | -0.002 |
| TTB@100 | -0.119 | -0.323 | -0.319 | -0.321 | -0.317 | -0.381 | -0.309 | -0.145 | 0.099 |
| RANK@1 | -0.026 | -0.264 | -0.264 | -0.264 | -0.318 | -0.287 | -0.365 | -0.112 | -0.189 |
| RANK@10 | -0.037 | -0.190 | -0.184 | -0.189 | -0.026 | -0.199 | -0.209 | -0.049 | -0.156 |
| RANK@100 | -0.069 | -0.085 | -0.085 | -0.087 | -0.032 | -0.159 | -0.093 | -0.069 | -0.051 |

Table 16: Results of the Wilcoxon signed rank test comparing the rankings induced by the ranked-based vs scored-based strategies. The statistically significant differences ($p\_value < 0.05$) in each group are highlighted in blue.

| | Avg | LR | Lasso | Ridge | RF | GB | LGBM | BO | MCTS |
|---|---|---|---|---|---|---|---|---|---|
| NDCG@1 | 0.001 | 0.000 | 0.000 | 0.000 | 0.000 | 0.000 | 0.000 | 0.000 | 0.000 |
| NDCG@10 | 0.000 | 0.000 | 0.000 | 0.000 | 0.000 | 0.000 | 0.000 | 0.000 | 0.000 |
| NDCG@100 | 0.000 | 0.000 | 0.000 | 0.000 | 0.000 | 0.000 | 0.000 | 0.000 | 0.000 |
| MRR@1 | 0.004 | 0.000 | 0.000 | 0.000 | 0.001 | 0.000 | 0.000 | 0.000 | 0.039 |
| MRR@10 | 0.000 | 0.000 | 0.000 | 0.000 | 0.000 | 0.000 | 0.000 | 0.000 | 0.304 |
| MRR@100 | 0.000 | 0.000 | 0.000 | 0.000 | 0.000 | 0.000 | 0.000 | 0.002 | 0.517 |
| SCORE@1 | 0.000 | 0.000 | 0.000 | 0.000 | 0.000 | 0.000 | 0.000 | 0.000 | 0.000 |
| SCORE@10 | 0.000 | 0.000 | 0.000 | 0.000 | 0.899 | 0.000 | 0.000 | 0.012 | 0.000 |
| SCORE@100 | 0.813 | 0.004 | 0.005 | 0.004 | 0.686 | 0.000 | 0.003 | 0.697 | 0.001 |
| TTB@1 | 0.250 | 0.000 | 0.000 | 0.000 | 0.006 | 0.000 | 0.000 | 0.011 | 0.000 |
| TTB@10 | 0.000 | 0.000 | 0.000 | 0.000 | 0.000 | 0.000 | 0.000 | 0.000 | 0.000 |
| TTB@100 | 0.000 | 0.000 | 0.000 | 0.000 | 0.000 | 0.000 | 0.000 | 0.000 | 0.072 |
| AVG_RANK@1 | 0.230 | 0.000 | 0.000 | 0.000 | 0.000 | 0.000 | 0.000 | 0.001 | 0.000 |
| AVG_RANK@10 | 0.141 | 0.000 | 0.000 | 0.000 | 0.234 | 0.000 | 0.000 | 0.079 | 0.000 |
| AVG_RANK@100 | 0.015 | 0.006 | 0.006 | 0.005 | 0.178 | 0.000 | 0.003 | 0.016 | 0.063 |

## C.4 Correlation between metrics

Tables 17, 18 and 19 show all Spearman correlations between all the metrics calculated for each dataset, respectively.

Table 17: Correlation between metrics in OpenML-Weka dataset.

|  | N@1 | N@5 | N@10 | M@1 | M@5 | M@10 | S@1 | S@5 | S@10 | R@1 | R@5 | R@10 |
|---|---|---|---|---|---|---|---|---|---|---|---|---|
| N@1 | 1.00 | 0.87 | 0.72 | 0.57 | 0.57 | 0.55 | 0.88 | 0.55 | 0.86 | -0.41 | -0.27 | -0.56 |
| N@5 | 0.87 | 1.00 | 0.85 | 0.68 | 0.71 | 0.69 | 0.73 | 0.73 | 0.84 | -0.27 | -0.25 | -0.65 |
| N@10 | 0.72 | 0.85 | 1.00 | 0.81 | 0.81 | 0.78 | 0.55 | 0.86 | 0.80 | 0.01 | 0.05 | -0.39 |
| M@1 | 0.57 | 0.68 | 0.81 | 1.00 | 0.99 | 0.98 | 0.40 | 0.97 | 0.63 | 0.32 | 0.29 | -0.13 |
| M@5 | 0.57 | 0.71 | 0.81 | 0.99 | 1.00 | 0.99 | 0.37 | 0.97 | 0.66 | 0.28 | 0.21 | -0.21 |
| M@10 | 0.55 | 0.69 | 0.78 | 0.98 | 0.99 | 1.00 | 0.38 | 0.96 | 0.64 | 0.30 | 0.23 | -0.20 |
| S@1 | 0.88 | 0.73 | 0.55 | 0.40 | 0.37 | 0.38 | 1.00 | 0.35 | 0.74 | -0.60 | -0.43 | -0.61 |
| S@5 | 0.55 | 0.73 | 0.86 | 0.97 | 0.97 | 0.96 | 0.35 | 1.00 | 0.68 | 0.30 | 0.25 | -0.22 |
| S@10 | 0.86 | 0.84 | 0.80 | 0.63 | 0.66 | 0.64 | 0.74 | 0.68 | 1.00 | -0.36 | -0.28 | -0.74 |
| R@1 | -0.41 | -0.27 | 0.01 | 0.32 | 0.28 | 0.30 | -0.60 | 0.30 | -0.36 | 1.00 | 0.95 | 0.76 |
| R@5 | -0.27 | -0.25 | 0.05 | 0.29 | 0.21 | 0.23 | -0.43 | 0.25 | -0.28 | 0.95 | 1.00 | 0.78 |
| R@10 | -0.56 | -0.65 | -0.39 | -0.13 | -0.21 | -0.20 | -0.61 | -0.22 | -0.74 | 0.76 | 0.78 | 1.00 |

Table 18: Correlation between metrics in AMLB Classification dataset.

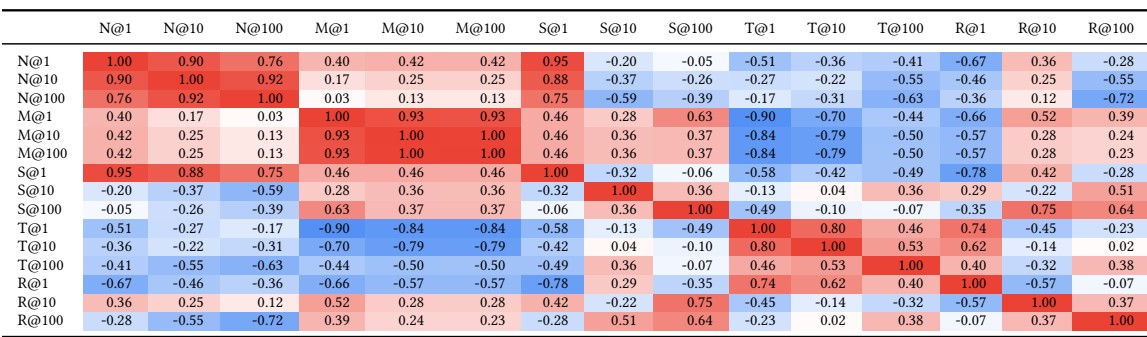

|  | N@1 | N@10 | N@100 | M@1 | M@10 | M@100 | S@1 | S@10 | S@100 | T@1 | T@10 | T@100 | R@1 | R@10 | R@100 |
|---|---|---|---|---|---|---|---|---|---|---|---|---|---|---|---|
| N@1 | 1.00 | 0.90 | 0.76 | 0.40 | 0.42 | 0.42 | 0.95 | -0.20 | -0.05 | -0.51 | -0.36 | -0.41 | -0.67 | 0.36 | -0.28 |
| N@10 | 0.90 | 1.00 | 0.92 | 0.17 | 0.25 | 0.25 | 0.88 | -0.37 | -0.26 | -0.27 | -0.22 | -0.55 | -0.46 | 0.25 | -0.55 |
| N@100 | 0.76 | 0.92 | 1.00 | 0.03 | 0.13 | 0.13 | 0.75 | -0.59 | -0.39 | -0.17 | -0.31 | -0.63 | -0.36 | 0.12 | -0.72 |
| M@1 | 0.40 | 0.17 | 0.03 | 1.00 | 0.93 | 0.93 | 0.46 | 0.28 | 0.63 | -0.90 | -0.70 | -0.44 | -0.66 | 0.52 | 0.39 |
| M@10 | 0.42 | 0.25 | 0.13 | 0.93 | 1.00 | 1.00 | 0.46 | 0.36 | 0.37 | -0.84 | -0.79 | -0.50 | -0.57 | 0.28 | 0.24 |
| M@100 | 0.42 | 0.25 | 0.13 | 0.93 | 1.00 | 1.00 | 0.46 | 0.36 | 0.37 | -0.84 | -0.79 | -0.50 | -0.57 | 0.28 | 0.23 |
| S@1 | 0.95 | 0.88 | 0.75 | 0.46 | 0.46 | 0.46 | 1.00 | -0.32 | -0.06 | -0.58 | -0.42 | -0.49 | -0.78 | 0.42 | -0.28 |
| S@10 | -0.20 | -0.37 | -0.59 | 0.28 | 0.36 | 0.36 | -0.32 | 1.00 | 0.36 | -0.13 | 0.04 | 0.36 | 0.29 | -0.22 | 0.51 |
| S@100 | -0.05 | -0.26 | -0.39 | 0.63 | 0.37 | 0.37 | -0.06 | 0.36 | 1.00 | -0.49 | -0.10 | -0.07 | -0.35 | 0.75 | 0.64 |
| T@1 | -0.51 | -0.27 | -0.17 | -0.90 | -0.84 | -0.84 | -0.58 | -0.13 | -0.49 | 1.00 | 0.80 | 0.46 | 0.74 | -0.45 | -0.23 |
| T@10 | -0.36 | -0.22 | -0.31 | -0.70 | -0.79 | -0.79 | -0.42 | 0.04 | -0.10 | 0.80 | 1.00 | 0.53 | 0.62 | -0.14 | 0.02 |
| T@100 | -0.41 | -0.55 | -0.63 | -0.44 | -0.50 | -0.50 | -0.49 | 0.36 | -0.07 | 0.46 | 0.53 | 1.00 | 0.40 | -0.32 | 0.38 |
| R@1 | -0.67 | -0.46 | -0.36 | -0.66 | -0.57 | -0.57 | -0.78 | 0.29 | -0.35 | 0.74 | 0.62 | 0.40 | 1.00 | -0.57 | -0.07 |
| R@10 | 0.36 | 0.25 | 0.12 | 0.52 | 0.28 | 0.28 | 0.42 | -0.22 | 0.75 | -0.45 | -0.14 | -0.32 | -0.57 | 1.00 | 0.37 |
| R@100 | -0.28 | -0.55 | -0.72 | 0.39 | 0.24 | 0.23 | -0.28 | 0.51 | 0.64 | -0.23 | 0.02 | 0.38 | -0.07 | 0.37 | 1.00 |

Table 19: Correlation between metrics in AMLB Regression dataset.

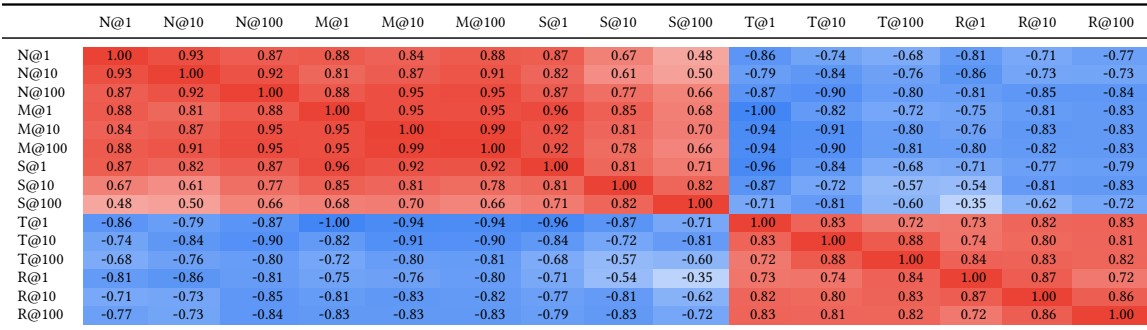

|  | N@1 | N@10 | N@100 | M@1 | M@10 | M@100 | S@1 | S@10 | S@100 | T@1 | T@10 | T@100 | R@1 | R@10 | R@100 |
|---|---|---|---|---|---|---|---|---|---|---|---|---|---|---|---|
| N@1 | 1.00 | 0.93 | 0.87 | 0.88 | 0.84 | 0.88 | 0.87 | 0.67 | 0.48 | -0.86 | -0.74 | -0.68 | -0.81 | -0.71 | -0.77 |
| N@10 | 0.93 | 1.00 | 0.92 | 0.81 | 0.87 | 0.91 | 0.82 | 0.61 | 0.50 | -0.79 | -0.84 | -0.76 | -0.86 | -0.73 | -0.73 |
| N@100 | 0.87 | 0.92 | 1.00 | 0.88 | 0.95 | 0.95 | 0.87 | 0.77 | 0.66 | -0.87 | -0.90 | -0.80 | -0.81 | -0.85 | -0.84 |
| M@1 | 0.88 | 0.81 | 0.88 | 1.00 | 0.95 | 0.95 | 0.96 | 0.85 | 0.68 | -1.00 | -0.82 | -0.72 | -0.75 | -0.81 | -0.83 |
| M@10 | 0.84 | 0.87 | 0.95 | 0.95 | 1.00 | 0.99 | 0.92 | 0.81 | 0.70 | -0.94 | -0.91 | -0.80 | -0.76 | -0.83 | -0.83 |
| M@100 | 0.88 | 0.91 | 0.95 | 0.95 | 0.99 | 1.00 | 0.92 | 0.78 | 0.66 | -0.94 | -0.90 | -0.81 | -0.80 | -0.82 | -0.83 |
| S@1 | 0.87 | 0.82 | 0.87 | 0.96 | 0.92 | 0.92 | 1.00 | 0.81 | 0.71 | -0.96 | -0.84 | -0.68 | -0.71 | -0.77 | -0.79 |
| S@10 | 0.67 | 0.61 | 0.77 | 0.85 | 0.81 | 0.78 | 0.81 | 1.00 | 0.82 | -0.87 | -0.72 | -0.57 | -0.54 | -0.81 | -0.83 |
| S@100 | 0.48 | 0.50 | 0.66 | 0.68 | 0.70 | 0.66 | 0.71 | 0.82 | 1.00 | -0.71 | -0.81 | -0.60 | -0.35 | -0.62 | -0.72 |
| T@1 | -0.86 | -0.79 | -0.87 | -1.00 | -0.94 | -0.94 | -0.96 | -0.87 | -0.71 | 1.00 | 0.83 | 0.72 | 0.73 | 0.82 | 0.83 |
| T@10 | -0.74 | -0.84 | -0.90 | -0.82 | -0.91 | -0.90 | -0.84 | -0.72 | -0.81 | 0.83 | 1.00 | 0.88 | 0.74 | 0.80 | 0.81 |
| T@100 | -0.68 | -0.76 | -0.80 | -0.72 | -0.80 | -0.81 | -0.68 | -0.57 | -0.60 | 0.72 | 0.88 | 1.00 | 0.84 | 0.83 | 0.82 |
| R@1 | -0.81 | -0.86 | -0.81 | -0.75 | -0.76 | -0.80 | -0.71 | -0.54 | -0.35 | 0.73 | 0.74 | 0.84 | 1.00 | 0.87 | 0.72 |
| R@10 | -0.71 | -0.73 | -0.85 | -0.81 | -0.83 | -0.82 | -0.77 | -0.81 | -0.62 | 0.82 | 0.80 | 0.83 | 0.87 | 1.00 | 0.86 |
| R@100 | -0.77 | -0.73 | -0.84 | -0.83 | -0.83 | -0.83 | -0.79 | -0.83 | -0.72 | 0.83 | 0.81 | 0.82 | 0.72 | 0.86 | 1.00 |

## C.5 Bonferroni Correction

Table 20 describes the experimental results using the Bonferroni correction for each metric, considering the number of independent tests as the product of the number of seeds, the number of tasks, and the number of models evaluated for the metric. The column "p-value" contains the p-value obtained after performing the Wilcoxon signed-rank test between the score-based and rank-based versions. The column "$\alpha_{\text{Bonferroni}}$" contains the corrected value (i.e., $\alpha/\#$ independent tests, with $\alpha = 0.05$). Finally, the column "<" indicates whether the p-value is less than $\alpha_{\text{Bonferroni}}$, meaning that the null hypothesis—that the metric value obtained by the rank-based version is less than or equal to that of the score-based version—is rejected.

Table 20: Results of the statistical significance tests for each metric after Bonferroni correction

|  | #independent tests ($seeds * tasks * models$) | p-value | $\alpha_{\text{Bonferroni}}$ | diff | < 0 |
|---|---|---|---|---|---|
| NDCG@1 | 17760 | 0.00e+0 | 2.82e-6 | -2.82e-6 | yes |
| NDCG@5 | 8400 | 7.56e-239 | 5.95e-6 | -5.95e-6 | yes |
| NDCG@10 | 17760 | 0.00e+0 | 2.82e-6 | -2.82e-6 | yes |
| NDCG@100 | 9360 | 0.00e+0 | 5.34e-6 | -5.34e-6 | yes |
| MRR@1 | 17760 | 1.51e-26 | 2.82e-6 | -2.82e-6 | yes |
| MRR@5 | 8400 | 1.21e-43 | 5.95e-6 | -5.95e-6 | yes |
| MRR@10 | 17760 | 3.07e-108 | 2.82e-6 | -2.82e-6 | yes |
| MRR@100 | 9360 | 6.12e-49 | 5.34e-6 | -5.34e-6 | yes |
| SCORE@1 | 17760 | 1.65e-303 | 2.82e-6 | -2.82e-6 | yes |
| SCORE@5 | 8400 | 3.22e-100 | 5.95e-6 | -5.95e-6 | yes |
| SCORE@10 | 17760 | 4.27e-30 | 2.82e-6 | -2.82e-6 | yes |
| SCORE@100 | 9360 | 4.13e-17 | 5.34e-6 | -5.34e-6 | yes |
| TTB@1 | 9360 | 1.07e-27 | 5.34e-6 | -5.34e-6 | yes |
| TTB@10 | 9360 | 1.72e-15 | 5.34e-6 | -5.34e-6 | yes |
| TTB@100 | 9360 | 4.77e-60 | 5.34e-6 | -5.34e-6 | yes |
| AVG_RANK@1 | 17760 | 3.65e-209 | 2.82e-6 | -2.82e-6 | yes |
| AVG_RANK@5 | 8400 | 3.13e-100 | 5.95e-6 | -5.95e-6 | yes |
| AVG_RANK@10 | 17760 | 1.49e-32 | 2.82e-6 | -2.82e-6 | yes |
| AVG_RANK@100 | 9360 | 2.63e-11 | 5.34e-6 | -5.34e-6 | yes |

## D  Pipeline Search Space

We describe the search space for machine learning pipelines as a grammar, with non-terminal symbols in ALL CAPS and terminal symbols in CamelCase. This encompasses the various components that can be assembled to preprocess data, select features, and construct models for classification or regression tasks. The structure of the pipeline is modular, allowing for different preprocessing techniques for categorical and numerical data, various feature selection methods, and a wide range of classifiers and regressors. A simplified grammar is described below.

PIPELINE := DATA_PREPROCESS & FEATURE_SELECTOR & (CLASSIFIER | REGRESSOR)

DATA_PREPROCESS := CATEGORICAL | NUMERICAL | (CATEGORICAL & NUMERICAL)

CATEGORICAL := (CategoricalImputation & ENCODING) | ENCODING

ENCODING := NoEncoding | OneHotEncoder

NUMERICAL := (NumericalImputation & SCALING) | SCALING

SCALING := NoRescaling | StandardScaler | MinMaxScaler |
Normalizer | QuantileTransformer | RobustScaler

FEATURE_SELECTOR := NoPreprocessing | Densifier | ExtraTreesPreprocessor |
FastICA | FeatureAgglomeration | KernelPCA |
RandomKitchenSinks | LibLinear | Nystroem |
PCA | PolynomialFeatures | RandomTreesEmbedding |
SelectPercentile | SelectClassificationRates | TruncatedSVD

CLASSIFIER := AdaboostClassifier | BernoulliNBClassifier |
DecisionTreeClassifier | ExtraTreesClassifier |
GaussianNBClassifier | GradientBoostingClassifier |
KNearestNeighborsClassifier | LDAClassifier |
LibLinear_SVCClassifier | LibSVM_SVCClassifier |
MultinomialNBClassifier | PassiveAggressiveClassifier |
QDAClassifier | RandomForestClassifier | SGDClassifier | MLPClassifier

REGRESSOR := AdaboostRegressor | ARDRegressor |
DecisionTreeRegressor | ExtraTreesRegressor |
GaussianProcessRegressor | GradientBoostingRegressor |
KNearestNeighborsRegressor | LibSVM_SVRegressor |
MLPRegressor | RandomForestRegressor | SGDRegressor

# E Simulated Experiment: Robustness of Rank-Based method

To complement our empirical analysis on real-world datasets, we designed a controlled simulation to evaluate the robustness of rank-based aggregation under increasing noise levels. This experiment aims to isolate the effects of noise on the ranking quality of two strategies: *score-based average* and *rank-based average*, when selecting pipelines across tasks.

**Setup.** We simulate a meta-learning scenario with $T = 200$ tasks and $C = 20$ configurations. For each task $t$, we sample a vector of ground-truth performance scores $s_t \in \mathbb{R}^C$ using a uniform distribution. To make the simulation realistic and heterogeneous, each task draws scores from a different range:

$$s_t^{(i)} \sim \mathcal{U}\left(0.5 + 0.2\sin(t + \text{seed}),\ 0.95 - 0.2\cos(t + \text{seed})\right).$$

This ensures variability in scale, spread, and ordering of configuration performance across tasks, mimicking what is typically observed in real-world meta-data.

We then introduce Gaussian noise to the scores:

$$\tilde{s}_t^{(i)} = s_t^{(i)} + \epsilon, \quad \epsilon \sim \mathcal{N}(0, \sigma),$$

for noise levels $\sigma \in \{0.01, 0.02, \ldots, 1.0\}$ (log-spaced). This models task-specific measurement noise, overfitting, or instability in reported performance.

For each noise level, and each seed in a set of 10 random seeds, we compute:

- **Score-Based Average**: For each configuration $c$, compute the average noisy score across tasks:

$$\bar{s}_c = \frac{1}{T} \sum_{t=1}^{T} \tilde{s}_t^{(c)}.$$

- **Rank-Based Average**: For each task, convert the noisy scores into inverse ranks (best = highest). Then compute the average inverse rank:

$$r_t^{(c)} = C - \text{rank}_t(c), \quad \bar{r}_c = \frac{1}{T} \sum_{t=1}^{T} r_t^{(c)},$$

where $C$ is the number of configurations.

**Evaluation.** We evaluate the quality of each global ranking using Kendall's Tau ($\tau$) between the predicted ranking (score-based or rank-based) and a reference ground truth. Importantly, the ground truth is defined as the ranking resulting from the true (noise-free) scores, ensuring that the evaluation focuses on preserving relative configuration quality.

**Results.** For each method and noise level, we average the Kendall's Tau over all seeds and report the standard deviation. Figure 5 shows that rank-based aggregation consistently achieves higher correlation with the ground-truth ranking under increasing noise, and demonstrates lower variance across seeds.

Finally, this controlled experiment shows that rank-based strategy is more robust to noise than score-based strategy, especially when performance scores are not directly comparable across tasks. These results align with our findings on real-world datasets, supporting the generalization capacity of rank-based methods for pipeline selection in meta-learning.

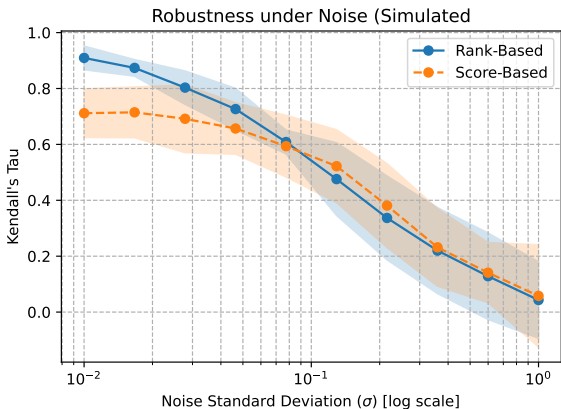

Figure 5: Kendall's Tau of predicted global rankings under increasing noise ($\sigma$) in simulated task performance data. Results are averaged over 5 seeds. The evaluation ground truth is based on average rank over noise-free scores.

