# OpenReview forum: "The Ranking Trick: A Simple and Robust Alternative to Score-Based Regression for AutoML"
_automl.cc/AutoML/2025/Methods_Track — AutoML 2025 Methods Track_

### Official Review · Reviewer_sQMu · 2025-04-30

**Comments To Authors:**

The paper presents a learning-to-rank (LTR) approach, as an alternative to the score-based (SB) approach, for ML pipeline selection. The presented approach, compatible with a wide range of AutoML techniques, shows superior performance to various baseline techniques compared on OpenML as well as synthetic data.

I have two concerns regarding the paper:

1) The description of the presented experiment is vague in some parts, for example:
- "We propose to do so by using ranking information instead of task-specific performance scores." --> based on what the ranking is made?
- "Meta-models were trained using 10-fold cross-validation on tasks and evaluated across 10 random seeds." --> Thus, 10 times 10-fold CV?
- "AVG_RANK (relative rank position between score- and ranking-based variants)." --> How exactly is this computed?
- "For each noise level, and each seed in a set of 10 random seeds, we compute" (Appendix) --> Need to clarify. Seeds are used in generating the scores but it is not clear why 10 seeds are used and how exactly. Also, for the first sight it is not clear that $i$ in $\tilde{s}_t^{(i)}$ refers to the number of configurations (maybe, stating that $0 < i \leq 20$ would help).
Using pseudocodes would help to resolve these vague parts.

2) Experiment results are, basically, just reported with only very brief attempts to justify the possible reasons. For example, why there are much less statistically significant differences in the results in the case of AMLB Classification and the AMLB Regression or the OpenML-Weka datasets in Table 2? I think that deeper discussion investigating possible causes for these phenomena would be a valuable addition to gain better insights to the presented approach.

Overall, the presented work shows promising directions for further research in the given area and is certainly interesting to the audience.

**Review Confidence:**

3

**Review Rating:**

7

---

### Official Review · Reviewer_rb1a · 2025-04-30

**Comments To Authors:**

***Summary Of Contributions***

This article presents a learning-to-rank (LTR) framework for solving the pipeline selection problem in automated machine learning (AutoML) systems.

***Potential Impact On The Field Of AutoML***

The use of ranks is not novel, but the experiment setup shows the difference between results and ranks, which benefits the community.
The impact is rather a practical guideline than an impact on AutoML research.

***Technical Quality And Correctness***

The work is well-written, although the appendix is disorganized (Table 3 - the best results are not highlighted in bold, Table 13 - Why are the SCORE values so high?). Some conceptions (e.g., the distinction between higher-best and lowest-best metric) are not well explained, but you can analyze the results. The included figures are clear and well-done (Figure 1, the metric name is missing).

Choose the way to write “learning-to-rank” as LtR or LTR.

***Overall Review***

*Positive*:

The paper is well structured.

Interesting results are relevant for practitioners.

*Negative*:

The appendix is disorganized.

The use of ranks is not novel.

**Review Confidence:**

4

**Review Rating:**

8

---

### Official Review · Reviewer_xNuG · 2025-05-09

**Comments To Authors:**

The paper presents a comprehensive benchmarking study comparing score-based and rank-based pipelines for algorithm selection, in which the authors evaluate five selection models under both paradigms. Experiments are conducted on three OpenML benchmarks—OPENML-WEKA-2017, encompassing 105 classification tasks and 30 pipelines; AMLB 2023 Classification, covering 71 tasks and 2,160 pipelines; and AMLB 2023 Regression, comprising 33 tasks and 1,485 pipelines. The results show that the rank‐based setting consistently outperforms the score‐based setting.

Originality:

The novelty of this work is rather limited, as rank-based approaches have already been explored extensively and exhibit their own drawbacks when absolute rankings are used. For instance, two algorithms might produce nearly identical performance metrics—differences that could be practically meaningful yet fail to reach statistical significance, especially in continuous black-box optimization. Imposing an absolute ordering in such cases masks these subtle distinctions. By contrast, relative ranking methods preserve small but important differences and have been the focus of prior investigations (Kostovska et al., 2023; Cenikj et al., 2024).

References:
- Kostovska, A., Jankovic, A., Vermetten, D., Džeroski, S., Eftimov, T., & Doerr, C. (2023, July). Comparing algorithm selection approaches on black-box optimization problems. In Proceedings of the Companion Conference on Genetic and Evolutionary Computation (pp. 495–498).
- Cenikj, G., Petelin, G., & Eftimov, T. (2024). A cross-benchmark examination of feature-based algorithm selector generalization in single-objective numerical optimization. Swarm and Evolutionary Computation, 87, 101534.

However, I appreciate your comprehensive experiments, which are the real strength of this paper. I recommend reframing it as a benchmarking study of score-based versus rank-based settings rather than presenting a novel rank-based framework, and noting that you plan to explore relative ranking methods in future work.

Clarity: The paper is well written; please also include a discussion of relative ranking methods in the related work section.

Pros: A comprehensive benchmarking study of score-based versus rank-based algorithm selection pipelines.

Cons: Limited novelty.

**Review Confidence:**

5

**Review Rating:**

7

---

### Meta-Review · Area_Chair_yaWZ · 2025-05-09

**Recommendation:** Accept
**Confidence:** 5

**Metareview:**

I suggest positioning the paper as a comparative benchmark of score-based versus rank-based approaches instead of introducing a new rank-based framework, and stating your intention to investigate relative ranking techniques in future work. Please consider this when preparing the camera-ready version.

Originality:
The use of rank‐based pipelines in algorithm selection is not novel—previous studies have explored both absolute and relative ranking methods. Consequently, the paper’s primary contribution lies more in its benchmarking scope than in methodological innovation.

Significance:
By systematically comparing score‐based and rank‐based settings across multiple large benchmarks, the work offers valuable practical guidance for AutoML practitioners. It also lays the groundwork for future investigations into relative ranking methods, pointing to promising research directions.

Clarity:
While the prose is smooth, the paper would benefit from:

-Standardizing terminology (e.g., choose either “LTR” or “LtR” consistently).

-Ensuring all figures and tables clearly label metrics and highlight best results.

Pros:
- Comprehensive evaluation across three large OpenML benchmarks.
- Clear demonstration that rank‐based pipelines often outperform score‐based ones.
- Findings offer practical guidelines for AutoML system designers.

Cons:
- Limited methodological novelty—rank‐based approaches have been studied extensively.
- Disorganized appendix with inconsistent tables and unexplained values.
- Key concepts (e.g., relative vs. absolute ranking) and metric conventions are not fully explained.